# A dynamic scale-mixture model of motion in natural scenes

**Jared M Salisbury, Stephanie E Palmer***

Department of Organismal Biology and Anatomy, Department of Physics, Physics Frontier Center for Living Systems, The University of Chicago, Chicago, United States

**Abstract** Some of the most important tasks of visual and motor systems involve estimating the motion of objects and tracking them over time. Such systems evolved to meet the behavioral needs of the organism in its natural environment and may therefore be adapted to the statistics of motion it is likely to encounter. By tracking the movement of individual points in movies of natural scenes, we begin to identify common properties of natural motion across scenes. As expected, objects in natural scenes move in a persistent fashion, with velocity correlations lasting hundreds of milliseconds. More subtly, but crucially, we find that the observed velocity distributions are heavy-tailed and can be modeled as a Gaussian scale-mixture. Extending this model to the time domain leads to a dynamic scale-mixture model, consisting of a Gaussian process multiplied by a positive scalar quantity with its own independent dynamics. Dynamic scaling of velocity arises naturally as a consequence of changes in object distance from the observer and may approximate the effects of changes in other parameters governing the motion in a given scene. This modeling and estimation framework has implications for the neurobiology of sensory and motor systems, which need to cope with these fluctuations in scale in order to represent motion efficiently and drive fast and accurate tracking behavior.

## Editor's evaluation

This paper tackles an important problem: the statistics of natural motion. The statistics of natural stimuli are in general highly structured, and the properties of that structure has guided understanding of sensory coding. This paper extends this analysis to natural motion. The authors first characterize the non-Gaussian properties of natural motion, and then introduce a simple Gaussian scale-mixture model that captures that behavior. The model is developed in a clear and convincing way and the results are compelling.

*For correspondence: sepalmer@uchicago.edu

## Introduction

One of the great triumphs of theoretical neuroscience has been the success of the efficient coding hypothesis (*Barlow, 1961*), which posits that sensory neural systems are adapted to the statistics of the organism's natural environment. The importance of this hypothesis lies in its power to explain structural features of the nervous system, such as the shapes of nonlinear response functions (*Laughlin, 1981*) and receptive fields of sensory neurons (*Srinivasan et al., 1982*; *van Hateren, 1992*; *Dan et al., 1996*; *Machens et al., 2004*; *Doi and Lewicki, 2014*), in terms of its function as an efficient information processing device. The success of this theory, particularly in vision, has sparked significant interest in measuring natural scene statistics (for a review, see *Simoncelli and Olshausen, 2001*) and has continued to yield important results, like the ubiquity of non-Gaussian, heavy-tailed statistics and related nonlinear forms of dependency among scene features (*Ruderman and Bialek, 1994*; *Schwartz and Simoncelli, 2001*; *Pitkow, 2010*; *Zoran and Weiss, 2012*).

The observation of heavy-tailed distributions in the natural world connects with the rich structure that the external environment presents to an organism's sensors, across a variety of sensory modalities. In any of these input streams, the brain has to pick out the relevant features in this rich input space that are most important for the organism's survival – to select what matters. Adapting to this kind of structure and maintaining an efficient representation of behaviorally relevant features in the world is a common feature of early sensory systems. Understanding how this is achieved, mechanistically, requires more than just the observation and quantification of heavy tails in natural scenes. To be able to understand how the brain represents this structure efficiently, we need to model it to shed light on potential ways the brain compresses this rich structure into an actionable internal signal.

Organisms are not passive sensory processors; they must also produce adaptive behavior in a complex and dynamic natural environment, where tasks like capturing prey (*Borghuis and Leonardo, 2015*; *Mischiati et al., 2015*; *Yoo et al., 2020*; *Shaw et al., 2023*), fleeing predators (*Gabbiani et al., 2002*; *Card and Dickinson, 2008*; *Muijres et al., 2014*), and navigating obstacles (*Muller et al., 2023*) are all critical to survival. These behaviors inevitably involve prediction (*Berry et al., 1999*; *Spering et al., 2011*; *Ben-Simon et al., 2012*; *Leonardo and Meister, 2013*) in order to compensate for substantial sensory and motor delays (*Franklin and Wolpert, 2011*). The basis for such predictive behavior must be statistical regularities in the environment, but little is known about the statistics of the inputs relevant to such behaviors.

As a step toward characterizing the statistics of behaviorally relevant quantities in natural scenes, we focus on a feature fundamental to many essential sensation-to-action programs – the motion of objects. Object motion relative to the observer drives oculomotor tracking (*Krauzlis and Lisberger, 1994*; *Hayhoe et al., 2012*) and is an essential part of many crucial behaviors, like prey capture (*Bianco et al., 2011*; *Hoy et al., 2016*; *Michaiel et al., 2020*). Specialized circuitry as early as the retina distinguishes between object and background motion (*Ölveczky et al., 2003*; *Baccus et al., 2008*), while entire brain regions in the visual cortex of primates specialize in processing motion (*Maunsell and Van Essen, 1983*), with increasing complexity along the dorsal stream (*Mineault et al., 2012*).

While previous work has characterized motion in certain cases, often focusing on optical flow due to ego-motion (*Calow and Lappe, 2007*; *Roth and Black, 2007*; *Muller et al., 2023*), little is known about the statistics of object motion in the natural world. To address this, we analyze movies from the Chicago Motion Database (https://cmd.rcc.uchicago.edu/), which were shot and curated for the purposes of statistical analysis and for use as stimuli for neural recordings and visual psychophysics. Rather than trying to track discrete objects (which may be difficult even to define for some movies, like those of flowing water), we simplify the problem by tracking individual points within the image using classic techniques from computer vision (*Lucas and Kanade, 1981*; *Tomasi and Kanade, 1991*).

Given a point trajectory, the velocity along that trajectory is a spatially local description of an object's motion through three-dimensional space, projected onto the two-dimensional surface of a sensor array, such as a retina or camera. We find that point velocity is highly correlated on the subsecond timescale we measure, and therefore point trajectories are highly predictable. More subtly, the distributions of velocity along trajectories exhibit heavy tails and nonlinear dependencies, both across horizontal and vertical velocity components and across time. This suggests the presence of an underlying *scale* variable, or local standard deviation, so the local velocity can be modeled as a Gaussian scale-mixture (GSM) (*Andrews and Mallows, 1974*). These models were developed in previous work examining the responses of filters applied to natural images and sounds (*Wainwright et al., 2001*; *Schwartz and Simoncelli, 2001*). We find that the scale fluctuates within individual trajectories on a relatively short timescale, so it is an essential part of our description of natural motion. Despite considerable differences in the velocity statistics across movies, the dynamic scale-mixture structure is remarkably consistent. This has important implications both for the efficient encoding of motion signals by neurons – which must adapt to the fluctuating scale to make full use of their limited dynamic range (*Fairhall et al., 2001*; *Olveczky et al., 2007*; *Liu et al., 2016*) – and for behaviors relying on predictive tracking – which must take into account the highly non-Gaussian statistics of natural motion (*Ho and Lee, 1964*).

## Results

In order to build up a statistical description of motion in natural scenes, we analyze movies from the Chicago Motion Database, which consists of a variety of movies collected for statistical analysis and

for use as visual stimuli in experiments. All movies were recorded using a fixed camera, with scenes chosen to contain consistent, dense motion within the field of view for minutes at a time. Scenes include flowing water, plants moving in the wind, and groups of animals such as insects and fish. While natural visual input is dominated by the global optical flow during eye and head movements (*Muller et al., 2023*), object motion warrants specific attention because it is highly behaviorally relevant for essential behaviors like escape or prey capture. Note that these global and local motion signals are approximately additive, so one can combine them to form a more complete description of motion for a given organism. We analyze a total of 15 scenes, with a resolution of $512 \times 512$ pixels, each $2^{14}$ = 16,384 frames long at a frame rate of 60 Hz (~4.5 min). The high resolution, frame rate, and lack of compression of these movies are essential for getting precise motion estimates.

For each scene, we quantify local motion using a standard point tracking algorithm (*Lucas and Kanade, 1981*; *Tomasi and Kanade, 1991*). A set of tracking points is seeded randomly on each frame, then tracked both forward and backward in time to generate trajectories (see *Materials and methods* for details). Early visual and sensorimotor systems operate on a timescale of tens to hundreds

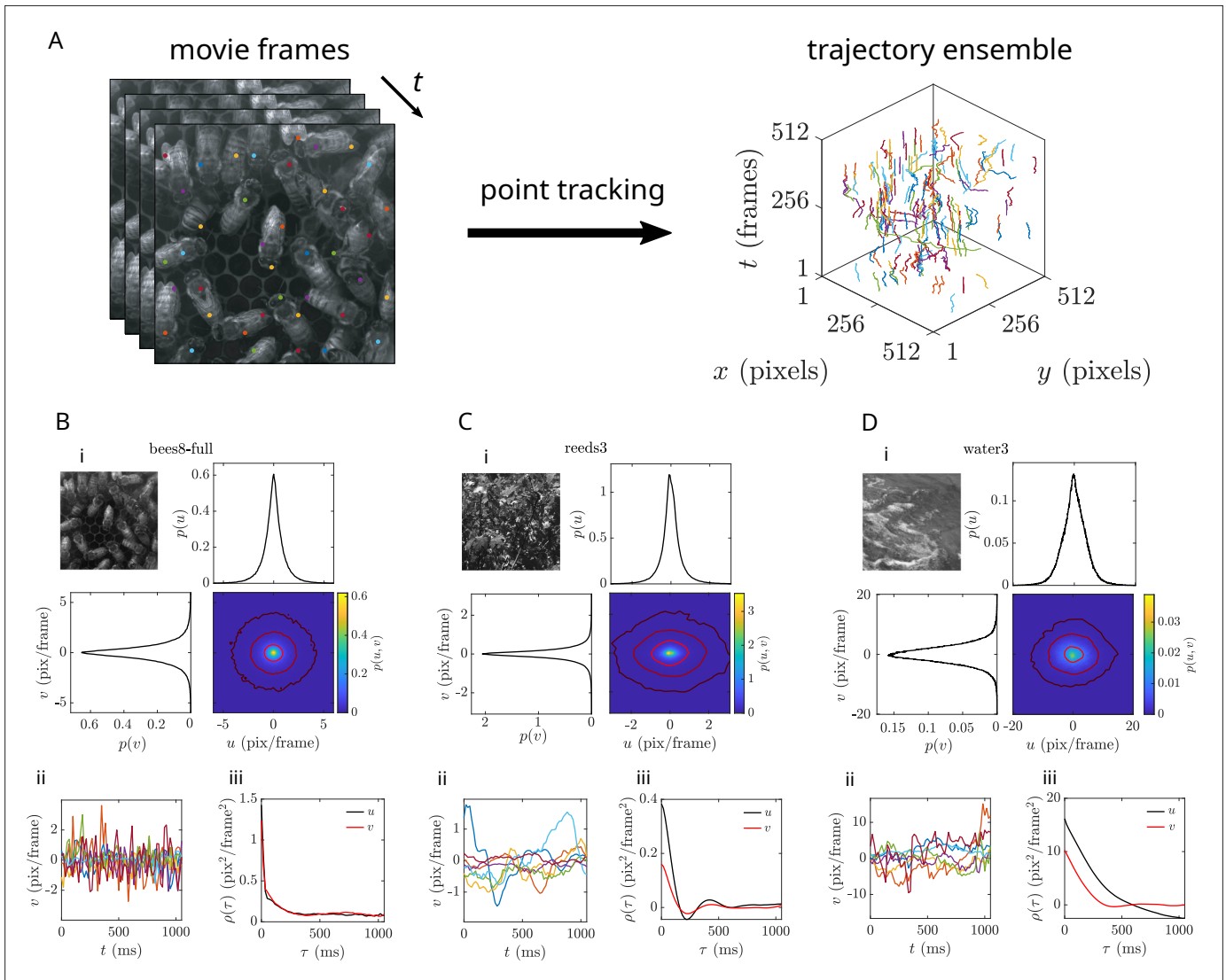

**Figure 1.** Automated point tracking reveals a diversity of motion statistics across natural scenes. (**A**) Natural movie data analyzed via point tracking yields an ensemble of ~1-s-long point trajectories. (**B–D**) Raw data summaries for three example movies, (**B**) `bees8-full`, (**C**) `trees14-1`, and (**D**) `water3`. (i) Joint and marginal distributions for horizontal (*u*) and vertical (*v*) velocity components. Overlaid isoprobability contours for the joint distributions are $p(u, v) = 10^{-1}, 10^{-2}$, and $10^{-3}$ for **B** and **C** and $p(u, v) = 10^{-2}, 10^{-3}$, and $10^{-4}$ for **D**. (ii) Seven example horizontal velocity component time series. (iii) Horizontal and vertical velocity correlation functions.

of milliseconds, so we restrict our analysis to short trajectories (64 frames, or ~1 s long) to reduce the amount of inevitable slippage from the point tracking algorithm. The resulting ensembles ($2^{13}$ = 8192 trajectories each) sparsely cover most of the moving objects in each movie (*Figure 1A*).

The focus of our analysis is the point velocity or difference in point positions between subsequent frames: a two-dimensional vector quantity measured in raw units of pixels/frame (this is easily converted to degrees of visual angle per unit of time, given a fixed viewing distance). The key advantage of this analysis is that the velocities are associated in time along a given point trajectory, which cannot be achieved by looking at the optical flow (*Horn and Schunck, 1981*) or motion energy (*Adelson and Bergen, 1985*) alone. Note that since tracking is a difficult problem, the distribution of velocity constrained to the ensemble of trajectories differs from the overall distribution, leading to underestimation of variance and kurtosis (see *Appendix 2*). This analysis is also distinct from previous work examining the spatiotemporal power spectra of natural scenes (*Dong and Atick, 1995*; *Billock et al., 2001*), since power spectra measure the globally averaged pairwise correlations between pixels.

Our understanding of motion in natural scenes must be grounded in what is perhaps the first scientific study of motion in a natural setting: the diffusive motion of pollen particles in water observed by *Brown, 1828*, later described theoretically by *Einstein, 1905*, and *Langevin, 1908*. See *Appendix 1* for a discussion of Brownian motion and its relation to our modeling framework. Briefly, Brownian motion is characterized by a Gaussian velocity distribution with an exponential correlation function.

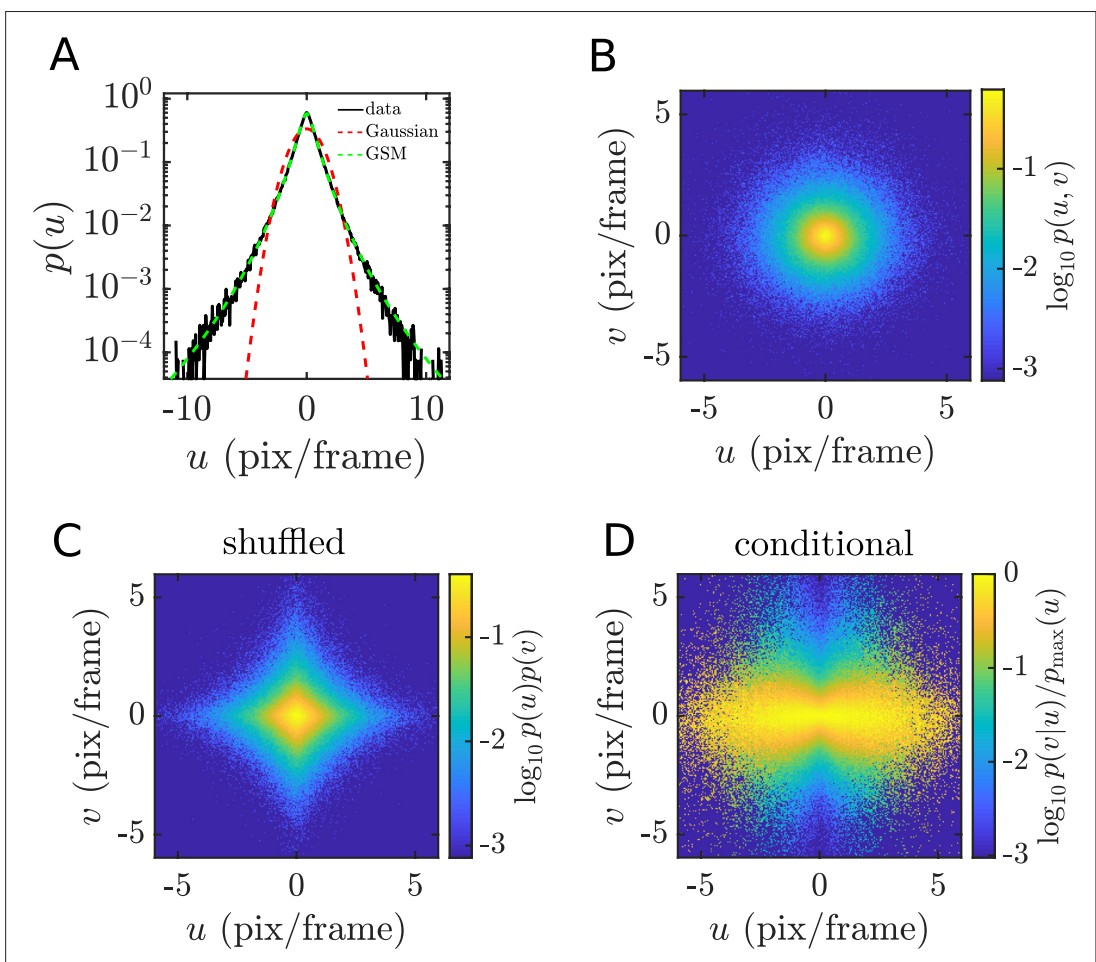

**Figure 2.** Velocity distributions are jointly heavy-tailed. (**A–D**) Velocity distributions for a single example movie, `bees8-full`. The marginal distribution for horizontal velocity (**A**) has much heavier tails than a Gaussian with the same variance and is well fit by a Gaussian scale-mixture model. The joint velocity distribution (**B**) is roughly radially symmetric, which differs substantially from the shuffled (and thereby, independent) distribution (**C**) and indicates a nonlinear dependence between the two velocity components. This dependence is alternatively revealed by the conditional distribution of the vertical velocity given the horizontal velocity (**D**), showing a characteristic bow-tie shape.

Natural scenes are, by definition, as richly varied as the natural world itself; each movie we analyze captures a small slice of this immense diversity. Our selection can be divided into three broad categories – animals, plants (animated by wind), and water – and we present summaries of the raw data for a representative movie from each category in *Figure 1B–D*. In contrast to the Gaussian velocity distributions expected for Brownian motion, histograms of the raw velocity data tend to be sharply peaked with long tails (see *Figure 2A* for a log-scale plot that emphasizes the tail behavior). Furthermore, the velocity time series exhibit correlation functions with diverse shapes, rather than simple exponential decay.

## Heavy-tailed statistics of natural motion

To examine the heavy-tailed structure of the observed point-trajectory velocity distributions, we pool horizontal and vertical velocity components together for an example movie, `bees8-full`, and compare this histogram to a Gaussian distribution with the same variance (*Figure 2A*). Plotted on a log-scale to highlight the tails, the empirical frequency falls off nearly linearly away from zero, while the Gaussian probability falls off quadratically. The same is true for the other movies in our dataset, pooled by category and all together (Figure 4A–C). Velocity distributions from animal and plant movies tend to have heavier tails, while those of water movies are closer to Gaussian.

When multiple variables are involved, heavy tails may be associated with a nonlinear form of dependency, as observed in the spatial structure of natural images (*Wainwright et al., 2001*). The same is true for the two velocity components in our data. We illustrate this for `bees8-full`, but results are similar for all other movies. In *Figure 2B*, we show a heat map of the joint histogram of horizontal and vertical velocities, $u$ and $v$. It is nearly radially symmetric. (For other movies with unequal variance in the two components, distributions are elliptic.) The lack of tilt in the histogram indicates that the two velocity components are uncorrelated (in this context, correlation between velocity components would indicate a tendency for objects within a scene to move along some diagonal axis relative to the camera). However, they are far from independent: when we shuffle the data to break any association between $u$ and $v$, the resulting histogram is no longer radially symmetric but is instead diamond-shaped (*Figure 2C*). This is, in fact, to be expected, since only Gaussian distributions may be both radially symmetric and separable. We demonstrate this dependency more clearly by plotting the conditional distribution of $v$ for each value of $u$, normalizing by the peak value at each $u$ for visualization purposes (*Figure 2D*). The resulting 'bow-tie' shape (*Schwartz and Simoncelli, 2001*) indicates that the variance of $v$ conditioned on $u$ increases with the magnitude of $u$. A simple, quantitative indicator of this nonlinear dependence is the Pearson correlation of the component magnitudes ($\rho = 0.33$, $p \ll 0.01$), which will be zero for any pair of independent random variables.

The form of the velocity distributions observed above suggests that they can be modeled as GSM distributions. As the name suggests, a GSM distribution is obtained by combining (zero-mean) Gaussian distributions of different scales, parameterized by a positive scalar random variable $S$. Let $Y$ be a Gaussian random variable with mean zero and variance $\sigma_Y^2$. If $S$ is a known quantity $s$, then $X = Ys$ is simply a Gaussian random variable with mean zero and variance $s^2\sigma_Y^2$. The conditional distribution is given by

$$p(x|s) = \mathcal{N}\left(x; 0, s^2\sigma_y^2\right)$$

When $S$ is unknown, $X = YS$ follows a GSM distribution given by

$$p(x) = \int_0^\infty p(x|s)p(s)ds,$$

where $p(s)$ is a distribution with positive support. A convenient choice is to let $S = \exp(Z)$, where $Z$ is Gaussian random variable, which we will refer to as the scale generator, with mean zero and variance $\sigma_Z^2$. Then, $S$ follows a log-normal distribution, which simplifies the inference problem significantly, despite the fact that the resulting GSM distribution does not have a closed form. The choice of a log-normal distribution can also be justified by a maximum entropy argument (*Van der Straeten and Beck, 2008*). See *Wainwright and Simoncelli, 2000*; *Wainwright et al., 2001*, for a discussion of the GSM model in the context of wavelet analysis of natural images. Parameters were estimated using a

variant of the expectation-maximization (EM) algorithm (*Dempster et al., 1977*) (see *Materials and methods*).

For an individual velocity component as in *Figure 2A*, the GSM model captures the shape of the distribution well, with only two parameters: $\sigma_Y$, controlling the overall scale, and $\sigma_Z$, controlling the heaviness of the tail. The variance of $X$ is related to these parameters by

$$\sigma_X^2 = \sigma_Y^2 \exp\left(2\sigma_Z^2\right).$$

The kurtosis, which is the standard measurement of tail heaviness, depends only on $\sigma_Z$:

$$\kappa_X = 3\exp\left(4\sigma_Z^2\right).$$

The kurtosis of $X$ thus grows exponentially with the variance of $Z$ and matches the Gaussian kurtosis of 3 if and only if $\sigma_Z = 0$.

To model the joint distribution, as in *Figure 2B*, clearly we cannot use independent GSM models for each component, since this corresponds to the shuffled distribution in *Figure 2C*. Instead, we consider a model in which two independent Gaussian random variables, $Y_1$ and $Y_2$, are multiplied by a shared scale variable $S$:

$$X_1 = Y_1 S,$$
$$X_2 = Y_2 S.$$

Note that we will maintain the general notation for the model for clarity. Applied to the velocity data, we have

$$(X_1, X_2) = (U, V),$$

and $(Y_1, Y_2)$ are the corresponding scale-normalized velocity components. The model is depicted in *Figure 3*, and it captures the radially symmetric (or more generally, when the variances are not equal, elliptic) shape of the joint distribution. Note that this model reveals why the Pearson correlation of the component magnitudes captures the nonlinear dependence between $U$ and $V$. We have, for the covariance (the numerator of the correlation coefficient),

$$E\left[(|X_1| - E[|X_1|])(|X_2| - E[|X_2|])\right] = E[|Y_1|]\,E[|Y_2|]\left(E[S^2] - E[S]^2\right),$$

which is positive whenever $S$ has nonzero variance.

This model has only three parameters: $\sigma_{Y_1}$, $\sigma_{Y_2}$, and $\sigma_Z$. (A more complicated model could have two correlated scale variables with different standard deviations. This does not appear to be necessary since the scale generator standard deviations fit independently to each component are nearly identical for most movies, and the elliptic shapes of the distributions indicate that the scale correlations across components are near one.) We observe a wide range of scale generator standard deviations $\sigma_Z$ both within and across categories (*Figure 4D*). The trend across categories – namely that animal and plant movies tend to have higher scale standard deviations than water movies – agrees with the relative heaviness of the tails for the pooled data (*Figure 4C*). On the other hand, $\sigma_{Y_1}$ and $\sigma_{Y_2}$ need not be similar, and many movies had a larger standard deviation of motion on the horizontal axis than vertical (the ratio $\sigma_{Y_2}/\sigma_{Y_1}$ tended to be less than one, *Figure 4E*). The AIC calculated for the two-dimensional GSM model with a common scale variable indicates that it is a better fit to the data than a two-dimensional, independent Gaussian model (*Figure 4F*).

## Coding implications of heavy tails

Heavy-tailed velocity distributions pose a particular challenge for efficient coding via sensory neurons. Consider the classic information theoretic problem of coding a random variable $X$ with an additive white Gaussian noise (AWGN) channel (*Cover and Thomas, 2006*) (in a more biologically realistic setting, one could consider, e.g., a population of Poisson neurons tuned to different motion directions, but the AWGN channel suffices for developing intuition). In this problem, the input is encoded with some function $f$, Gaussian noise $N$ is added by the channel, and the noise-corrupted signal is

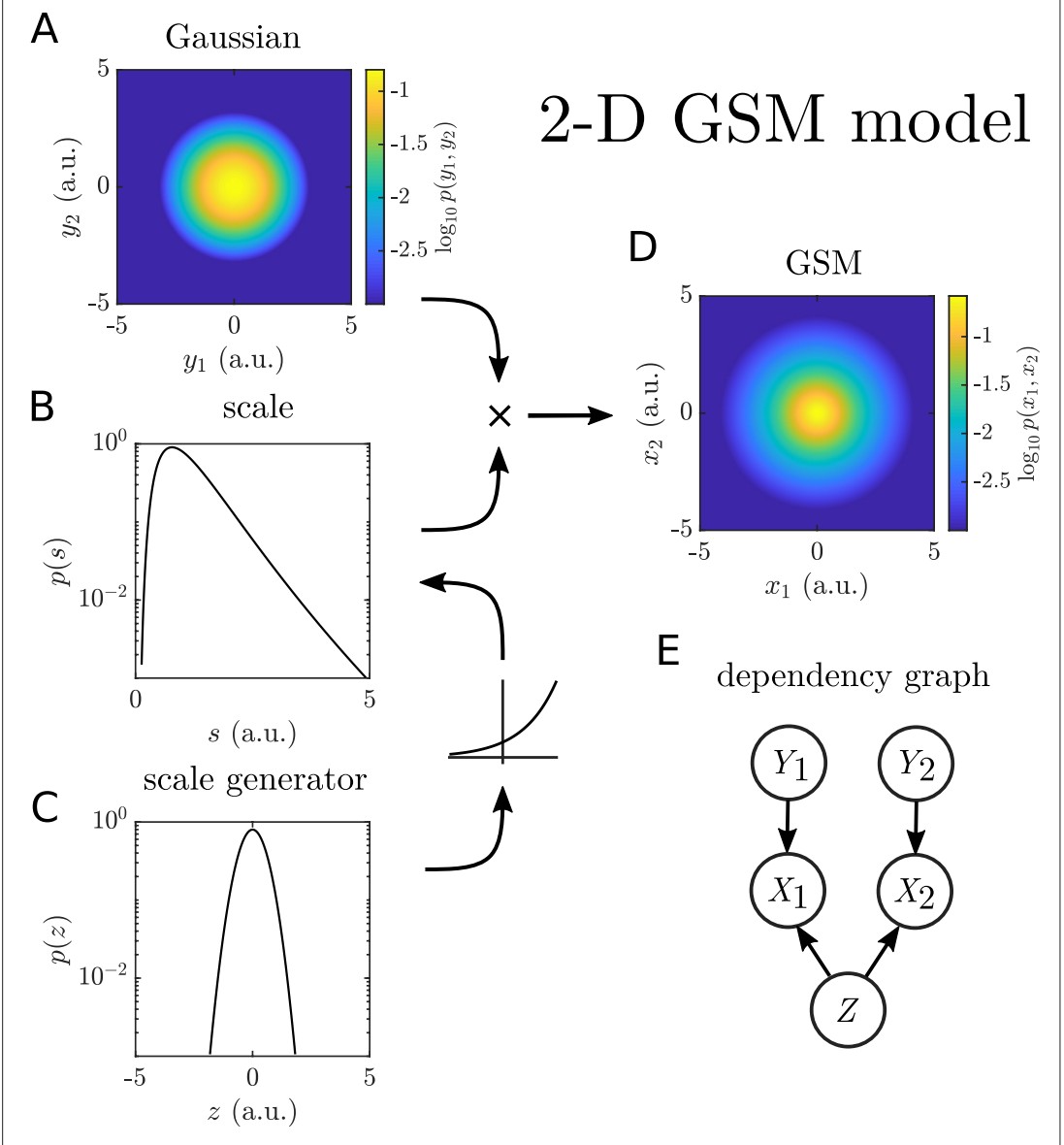

**Figure 3.** Schematic of the two-dimensional Gaussian scale-mixture model. (**A–D**) A Gaussian random variable $Z$ (**C**) is passed through an exponential nonlinearity to yield a log-normal scale variable $S$ (**B**). The scale multiplies both components of an underlying Gaussian distribution (**A**) to produce radially symmetric heavy tails (**D**). For the joint distributions, probabilities less than $10^{-3}$ were set to zero to facilitate comparison with empirical histograms. (**E**) Dependency graph for the variables in the model.

decoded by some function $g$ to form an estimate of the input, $\hat{X} = g\left[f(X) + N\right]$. The channel capacity, $C$, is a function of the signal-to-noise ratio (SNR),

$$C = \frac{1}{2}\log(1 + \text{SNR}),$$

where $\text{SNR} = \sigma^2_{f(X)}/\sigma^2_N$ is the ratio of the encoded signal variance to the noise variance. The mutual information $I(X,\hat{X})$ is equal to $C$ if and only if $f(X)$ is Gaussian. Otherwise, $I < C$, and the coding efficiency $E = I/C$ is less than one. It is commonly assumed that biological systems must expend energy to achieve a desired SNR, either by amplifying the signal or suppressing the noise, so that coding efficiency corresponds directly to energy efficiency.

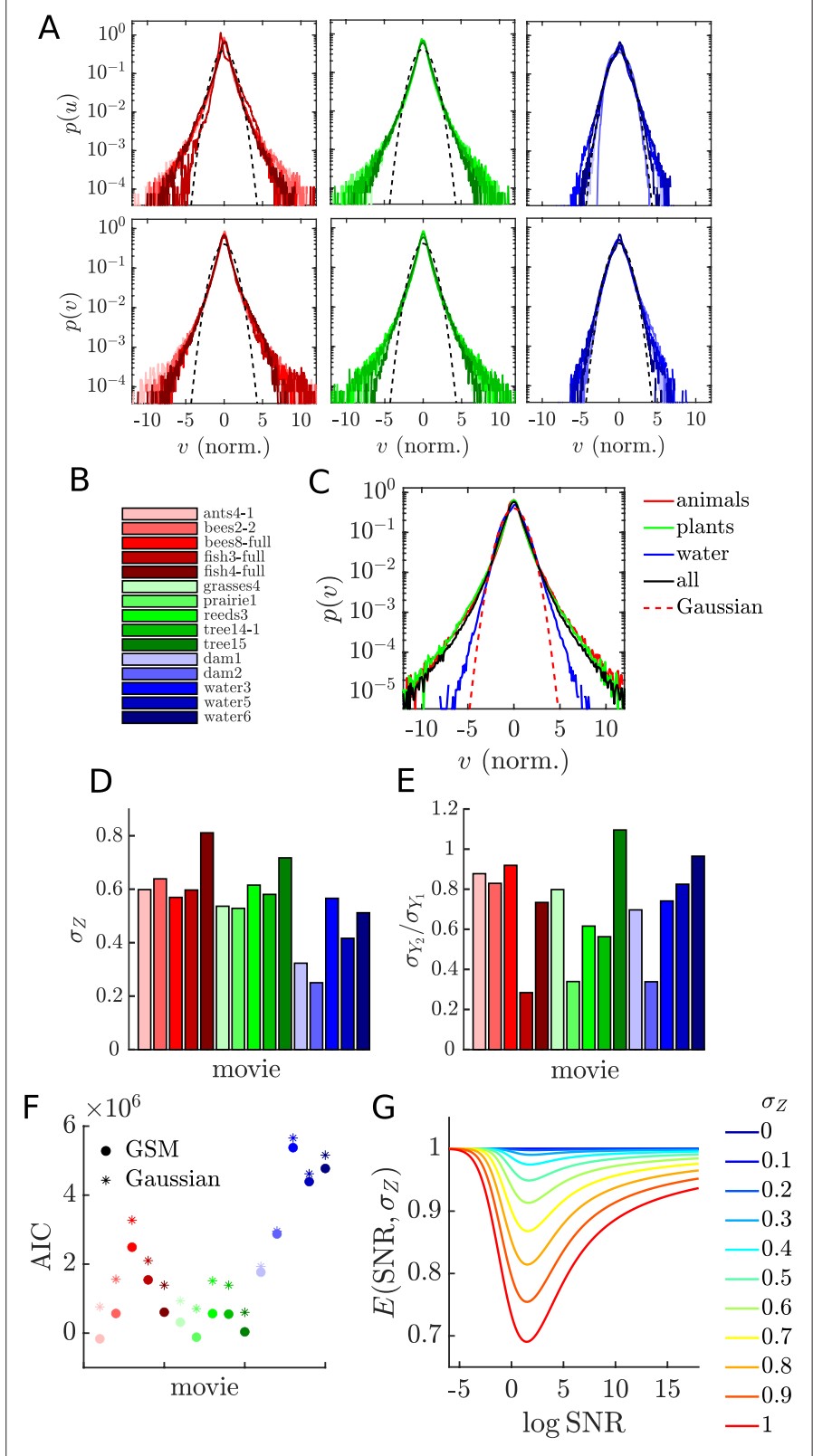

**Figure 4.** Quantifying heavy tails across scenes and categories (**A**). Marginal distributions for vertical and horizontal velocity components, grouped by category. (**B**) Legend of individual movie names for A and all subsequent plots. (**C**) Marginal distributions for the combined data across categories. Each velocity component of each movie was normalized by its standard deviation before combining. (**D**) Estimated standard deviations for the scale generator

*Figure 4 continued on next page*

*Figure 4 continued*

variable, $Z$, varied across movies, corresponding to different amounts of kurtosis. (**E**) The ratio of estimated standard deviations of the underlying Gaussian variables, $Y_1$ and $Y_2$, showing the degree of anisotropy. (**F**) Akaike information criterion (AIC) values for the two-dimensional, shared scale Gaussian scale-mixture (GSM) model versus the two-dimensional, independent Gaussian model. (**G**) Coding efficiency as a function of signal-to-noise ratio for different values of $\sigma_Z$.

---

When the encoding function is identity, $f(X) = X$, heavy tails in $p(x)$ lead to a loss of efficiency. We calculate the coding efficiency as a function of the parameters of the GSM model for $X$ and the noise level $\sigma_N^2$. We have

$$I(\hat{X}, X) = H(\hat{X}) - H(\hat{X}|X) = H(\hat{X}) - H(N),$$

where

$$H(\hat{X}) = -\int_{-\infty}^{\infty} p(\hat{x}) \log p(\hat{x}) d\hat{x},$$

$$p(\hat{x}) = \int_0^{\infty} p(\hat{x}|s)p(s)ds,$$

$$p(\hat{x}|s) = \mathcal{N}\left(\hat{x}; 0, s^2\sigma_Y^2 + \sigma_N^2\right),$$

and

$$H(N) = \frac{1}{2}\log\left(2\pi e \sigma_N^2\right).$$

To see the effect of heavy tails, we vary $\sigma_Z^2$ and the SNR, keeping either $\sigma_Y^2$ or $\sigma_N^2$ fixed, and calculate the above mutual information numerically (*Figure 4G*). Since $I$ and $C$ have the same scaling behavior with SNR, $E \to 1$ as $\log(\text{SNR}) \to \pm\infty$, the heavy tails have no effect at very high or low SNR. At intermediate SNR, the coding efficiency decreases monotonically as $\sigma_Z^2$ increases. The efficiency reaches a minimum at $\log \text{SNR} = 3/2$ for all $\sigma_Z^2 > 0$.

How can one encode $X$ for better efficiency? The most straightforward way is to 'Gaussianize' $X$ so that $f(X)$ is Gaussian (*Chen and Gopinath, 2000*), thereby achieving perfect efficiency. This can be done generically by choosing $f(X) = \Phi^{-1}\left[F(X)\right]$, where $F$ is the cumulative distribution function of $X$, and $\Phi$ is the cumulative distribution function of the standard Gaussian distribution. Neurons have been shown to implement this kind of efficient coding by matching their response nonlinearities to natural scene statistics (*Laughlin, 1981*), although the mapping is to a uniform distribution over a fixed interval rather than a Gaussian. (The uniform distribution is the maximum entropy distribution on an interval – here, the range of firing rates from zero to some upper limit – just as the Gaussian is the maximum entropy distribution on the real line with fixed variance. The following argument still applies in this setting.) In principle, this method can be applied to each time step and channel (velocity component) independently; however, if $X_1$ and $X_2$ are not independent, as demonstrated for velocity components above, the joint distribution of $f_1(X_1)$ and $f_2(X_2)$ will not be Gaussian, so that efficiency is still not perfect in terms of the two-dimensional generalization of the channel capacity. More complex encoding methods may yield approximately jointly Gaussian distributions (*Chen and Gopinath, 2000*; *Laparra et al., 2011*), but these are not biologically plausible. Another downside of this strategy is that it introduces signal-dependent noise to the decoded variable, since $\hat{X} = f^{-1}\left[f(X) + N\right] = X + \hat{N}(X)$. In particular, the velocity values with highest magnitude, which may be the most relevant for behavior, will have the highest noise, since $f$ must be compressive at the tails and therefore $f^{-1}$ must be expansive.

Another strategy is to demodulate or normalize $X$ by estimating $S$ and dividing $X$ by it. If the estimate is accurate, then $X/\hat{S} = \hat{Y} \approx Y$, a Gaussian, and channel efficiency for $\hat{Y}$ will be high. In order to recover $\hat{X}$, $\hat{Z} = \log \hat{S}$ will need to be encoded in another channel, and the two sources of additive noise will result in multiplicative noise and heavy-tailed additive noise in the estimate:

$$\hat{X} = (\hat{Y} + N_Y)\exp(\hat{Z} + N_Z) = X\exp(N_Z) + N_Y\exp(\hat{Z} + N_Z).$$

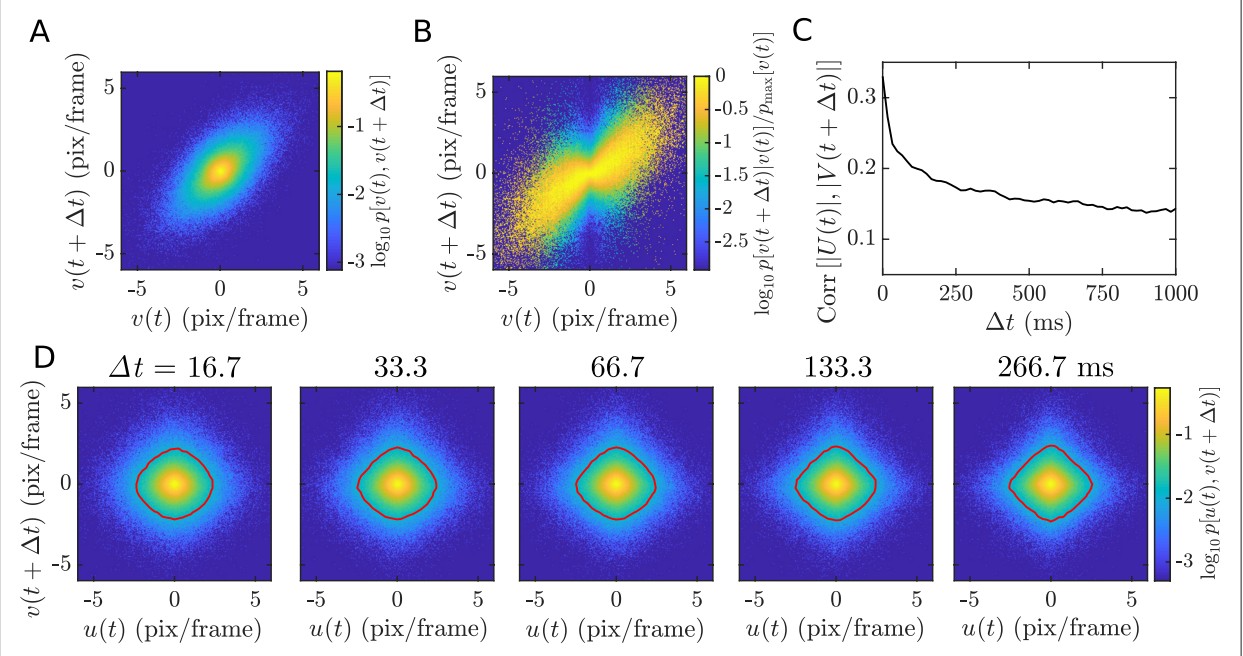

**Figure 5.** Temporal correlations in velocity and scale. (**A–B**) Joint (**A**) and conditional (**B**) histograms for horizontal velocity across two adjacent frames for an example movie (`bees8-full`). The tilt indicates a strong linear correlation, while the elliptic shape in (**A**) and bow-tie shape in (**B**) indicate the coexistence of a nonlinear dependence due to an underlying scale variable. (**C**) The correlation coefficient between velocity component magnitudes, offset in time by $\Delta t$, decays as a function of $\Delta t$, indicating that the shared scale variable fluctuates in time. (**D**) The joint distributions of the two components separated by $\Delta t$ show a gradual transformation from the original radially symmetric shape (**Figure 2B**) toward the diamond shape of the shuffled distribution (**Figure 2C**) (red curves are isoprobability contours at p=0.01).

Of course, this strategy fails for a single variable $X$ since the only reasonable estimate for the scale is $\hat{S} = |X|$, so that $\hat{Y} = \pm 1$. However, since the two velocity components share a common scale variable, the estimate can be improved by making use of both components, and these can be made approximately jointly Gaussian, unlike in the marginal Gaussianization strategy described above. Furthermore, since the scale may be correlated in time in natural scenes, as shown in the next section, the history of $X(t)$ can also be used to further improve the estimate $\hat{S}$. Calculating the overall cost of this strategy is complicated by the fact that it introduces a new channel for the scale variable with its own associated SNR-dependent energy consumption.

## The dynamics of natural motion

While the time-independent statistics of velocity are important, a full description of how objects move must include how the velocity evolves over time *along* point trajectories. (This description should ideally also include how multiple points on the same object evolve over time, allowing us to capture rotations, contractions, and expansions; we do not attempt this more ambitious analysis here and limit our discussion to local translations.) This motivates our point tracking analysis, which provides information that cannot be gleaned from motion estimates at fixed locations alone. From the raw data, we know the velocity is highly correlated at short time lags, but it is not clear how the scale variable enters into play. We again inspect the joint velocity distribution for an example movie, now across neighboring time points for one velocity component (**Figure 5A**). The tilt indicates strong linear correlation across time in the velocity, as expected, and we note that the overall shape is elliptic despite having heavy-tailed marginals, as in **Figure 2B**. In **Figure 5B**, we condition on the velocity at one time step and observe a tilted version of the same bow-tie shape as in **Figure 2D**. With the addition of an overall tilt, these are precisely the same observations that led us to the GSM model above. Thus, two forms of dependence – linear correlation and the nonlinear dependence due to the scale variable – coexist in the time domain.

We next ask whether this scale variable is constant in time (varying only from trajectory to trajectory) or dynamic (varying in time within a given trajectory). If it is constant, the joint distribution of

the two components will not depend on the alignment of the two components in time, so long as they are from the same trajectory. In **Figure 5D**, we examine these joint distributions after shifting one component relative to the other by a time lag, for a range of lags. The distributions gradually shift from the radially symmetric zero-lag distribution to a diamond shape similar to the shuffled distribution. Similarly, the correlation between velocity component magnitudes, an indicator of nonlinear dependence discussed above, decreases as the lag increases (**Figure 5C**). In other words, at zero lag, the two velocity components are governed by a single scale variable; at nonzero lags, they are governed by two correlated scale variables, and this correlation decreases with lag. This indicates that the underlying scale variable fluctuates over time. Notably, we can detect these fluctuations within the ~1-s-long trajectories to which we limit our analysis. This would not be the case if the scale were to change only on a very long timescale or only across different point trajectories within a scene.

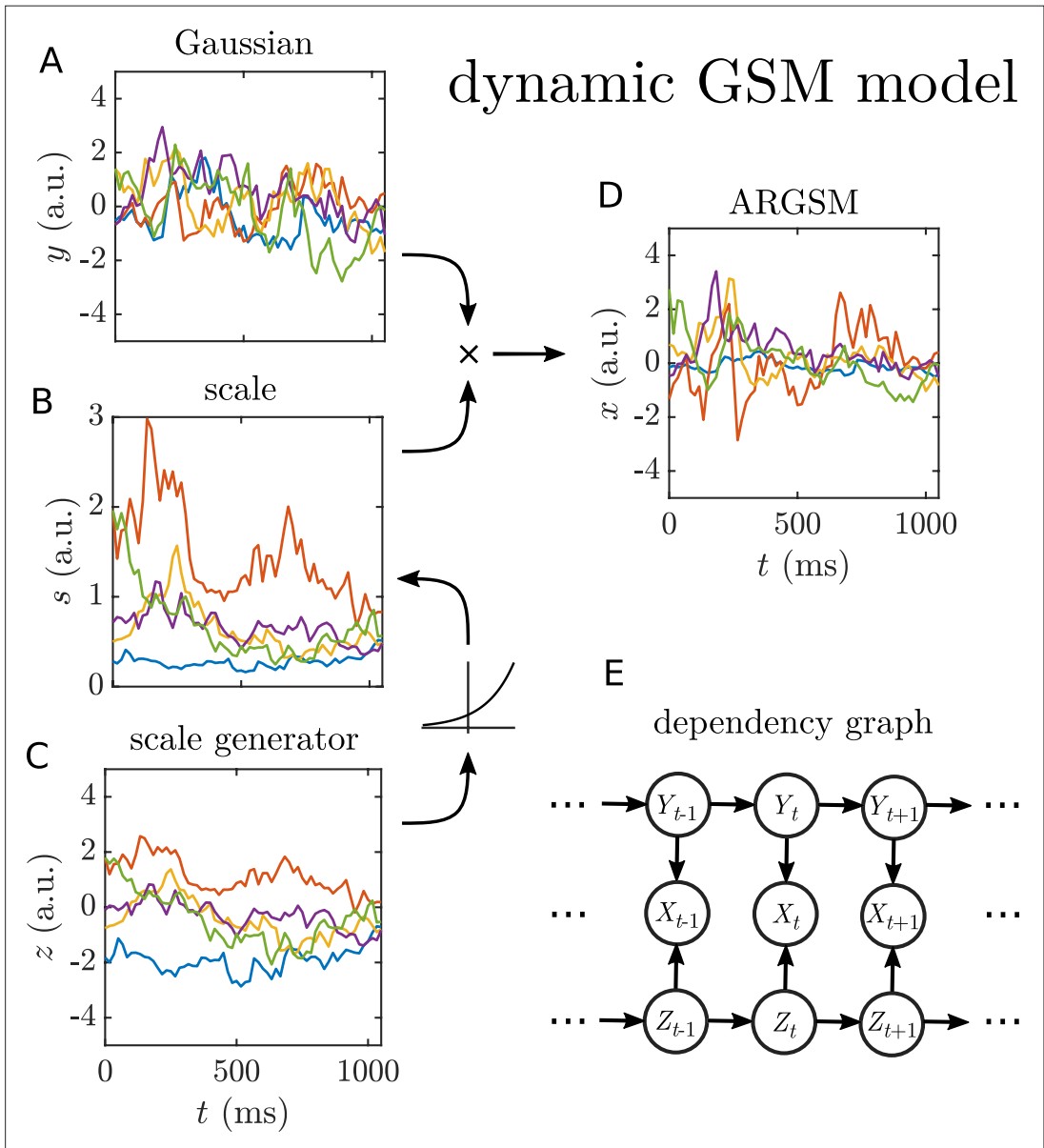

**Figure 6.** Schematic of the dynamic Gaussian scale-mixture model. (**A–D**) Both $Y$ (**A**) and $Z$ (**C**) are modeled by high-order autoregressive processes to capture arbitrary correlation functions. (Only AR(1) processes are depicted graphically and used to simulate data.) The scale process $S$ (**B**) is generated by passing $Z$ through an element-wise exponential nonlinearity. It then multiplies the underlying Gaussian process $Y$ element-wise to yield the observed process with fluctuating scale (**D**). Only one component is depicted. In the full model, two independent Gaussian processes share a common scale process. (**E**) Dependency graph for the variables in the one-dimensional model.

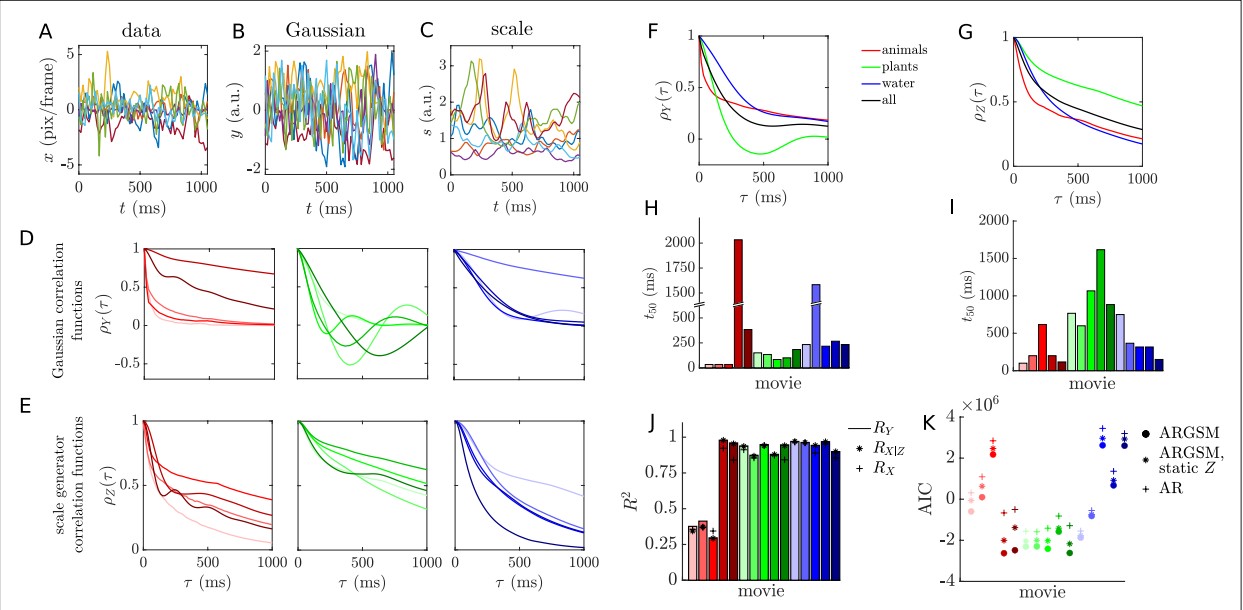

**Figure 7.** Quantifying velocity and scale correlations. (**A–C**) Example traces of the raw velocity (**A**), scale-normalized velocity (**B**), and estimated scale variable (**C**). (**D**) Temporal correlation functions for the underlying Gaussian processes of each movie, grouped by category. Horizontal and vertical components were averaged before normalizing (equivalently, each component was weighted by its variance). (**E**) As in **D**, for the scale-generating Gaussian process, $Z$. (**F**) The Gaussian process correlation functions in **D** averaged within categories. (**G**) As in F, for the scale-generating Gaussian process correlation functions in **E**. (**H**) Lag time to reach a correlation of 0.5 for the underlying velocity Gaussian processes for each movie (components were weighted by variance as in **D**). (**I**) As in **H**, for the scale-generating Gaussian process. (**J**) Predictive variance explained for each movie. Variances were averaged across horizontal and vertical components before calculating $R^2$. (**K**) Akaike information criterion (AIC) values for different models for each movie. Lower values indicate better model fit.

We would like to capture this dynamic scale variable in our model of natural motion. It is straight-forward to make the GSM model dynamic by replacing each Gaussian variable with an autoregressive Gaussian process, and we call this new model the ARGSM model. We illustrate it schematically in *Figure 6* by generating example traces for one Gaussian velocity component $Y$ and the scale gener-ator $Z$. Note that the autoregressive scale generator variable is the temporal equivalent to the spatial Markov random fields explored in the image domain (*Wainwright et al., 2001*; *Roth and Black, 2009*; *Lyu and Simoncelli, 2009*). Given this model, we perform estimation of the parameters using a stochastic approximation variant of the EM algorithm (see *Materials and methods*). (Code for fitting the ARGSM model is available at https://github.com/sepalmer/motion-scale-mixture, *Palmer and Salisbury, 2026*) Example traces illustrating the results of this model are shown in *Figure 7A–C*. The estimated autoregression coefficients determine the correlation functions of the underlying Gaussian velocity and scale generator processes, which we plot for each movie in *Figure 7D and E*, respec-tively. The fact that some velocity correlation functions and many scale generator correlation functions do not go to zero over the length of the trajectories could indicate a nonzero mean component that varies from trajectory to trajectory, but this is beyond the scope of the present analysis. Average correlation functions across categories are shown in *Figure 7F–G*. We also report the time to 0.5 correlation for each movie for the velocity in *Figure 7H–I*. Akaike information criterion (AIC) scores (*Figure 7K*) indicate that the full ARGSM is a better fit to the data compared to the AR model. It is also a better fit compared to the ARGSM model with a static $Z$ value for each trajectory, indicating that a dynamic scale variable is essential for describing the data. In the context of visual tracking of moving objects, the timescales of these correlation functions are extremely important. On one hand, the velocity correlation time determines how far into the future motion can be extrapolated. On the other hand, the scale correlation time determines the timescale on which adaptation must take place in order to efficiently process motion signals with limited dynamic range.

Finally, we ask whether the full ARGSM model is necessary to carry out scale normalization in practice for our trajectory data. Our model fitting provides an estimate of the scale at each time point, which we use to normalize the raw data. To quantify normalization performance, we calculate

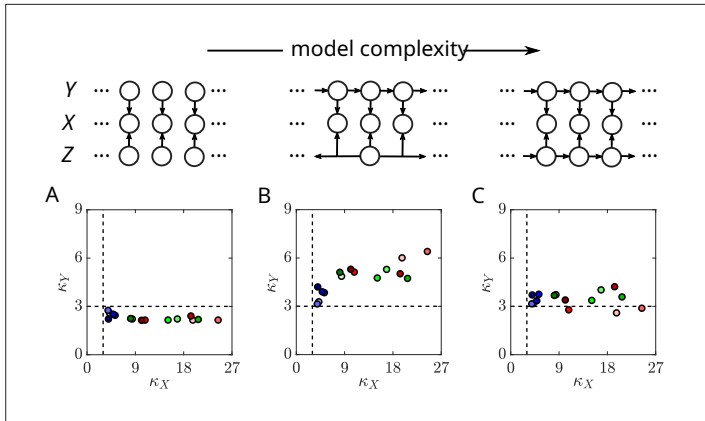

**Figure 8.** A dynamic scale-mixture model is necessary for effective normalization. (**A**) Kurtosis of the velocity before and after dividing by a point estimate of the scale (*bottom*) under the time-independent model (*top*). Kurtosis was computed by pooling the two components after normalizing by each standard deviation, so that differences in the variance across components do not contribute additional kurtosis. A Gaussian distribution has a kurtosis of 3 (dashed lines). (**B**) As in A, but for a model with autocorrelated Gaussian processes and a constant scale for each trajectory. (**C**) As in A, but for the fully dynamic model.

the kurtosis, or fourth standardized moment, which measures how heavy-tailed a distribution is. The standard reference is a Gaussian random variable, which has a kurtosis of 3. In *Figure 8*, we compare the kurtosis of the velocity before and after dividing by a point estimate of the scale under three models of increasing complexity. If normalization is successful, the distribution of the resulting normalized velocity should be approximately Gaussian. Under the time-independent model, the normalized velocity consistently has kurtosis less than 3, indicating that the scale tends to be overestimated (*Figure 8A*). In contrast, for a model with correlated velocity and constant scale for each trajectory, the kurtosis is consistently larger than 3, indicating that the scale tends to be underestimated (*Figure 8B*). Only the full model, with correlated velocity and a dynamic, correlated scale variable, yields a kurtosis around 3 for each movie, even with highly kurtotic data (*Figure 8C*).

This exercise of using the ARGSM model to estimate the scale at each time point, then dividing the velocity by this scale, serves as a proxy for what the nervous system can achieve through adaptation mechanisms. An important caveat is that the model has access to the full trajectory, while the nervous system must operate in an online, causal setting.

## Implications for prediction

Prediction is an important problem both for compression via predictive coding and for overcoming sensory and motor delays during behavior. Prediction is built into the ARGSM framework since the regression coefficients of the AR models are optimal for predicting the next time step of $Y_t$ and $Z_t$ given their past values. Let $\hat{Y}_t$ and $\hat{Z}_t$ denote the predicted values and $v_t$ and $\zeta_t$ the respective errors:

$$Y_t = \hat{Y}_t + v_t,$$

$$Z_t = \hat{Z}_t + \zeta_t.$$

The variance explained by a prediction $\hat{X}_t$ is given by

$$R^2 = \frac{E\left[X_t^2\right] - E\left[\left(X_t - \hat{X}_t\right)^2\right]}{E\left[X_t^2\right]}.$$

For the Gaussian process $Y_t$, this simplifies to

$$R_Y^2 = \frac{\sigma_{\hat{Y}}^2}{\sigma_Y^2} = 1 - \frac{\sigma_v^2}{\sigma_Y^2}.$$

Assuming knowledge of the histories of both $X_t$ and $Z_t$, the prediction for $X_t$ is

$$\hat{X}_t = \hat{Y}_t \exp(\hat{Z}_t),$$

The associated variance explained is

$$R^2_{X|Z} = R^2_Y \left( 2 \exp \frac{1}{2} \sigma^2_\zeta - \exp 2\sigma^2_\zeta \right).$$

This is an upper bound on the performance of any predictor with access only to $X_t$.

Notably, $R^2_{X|Z}$ is independent of $\sigma^2_Z$: the variance explained under the ARGSM model for $X$ is equal to the variance explained for $Y$, multiplied by a function of the variance of the driving noise for $Z$, $\sigma^2_\zeta$, that slowly decreases from one to zero. Since $\sigma^2_\zeta$ is small for the estimated models, indicating that the scale fluctuations are highly predictable at the level of single time steps, we expect this term to have little effect. In *Figure 7J*, we compare the variance explained by applying naive autoregression to $X_t$ (denoted $R^2_X$) to $R^2_Y$ and $R^2_{X|Z}$ using the estimated model parameters. The variance explained is close to one for all movies except three depicting insects (a consequence of their very short velocity correlation times). Values of $R^2_X$ tend to be only slightly smaller than $R^2_{X|Z}$ (except in two instances where it was slightly larger). We conclude that heavy-tailed statistics have little effect on the predictability of natural motion, although scale estimation is necessary for estimating the variance associated with the prediction, i.e., the variance of the posterior distribution $p\left(x_t | \overleftarrow{\mathbf{x}}_{t-1}\right) \approx \mathcal{N}\left(x_t; \hat{x}_t, \sigma^2_v \exp\left(2\hat{z}_t\right)\right)$.

Long correlation times and high values of $R^2$ indicate that the velocity time series of natural motion are highly predictable. One way to make use of this predictability is through predictive coding, in which only prediction errors (with variance $\sigma^2_X - \sigma^2_{\hat{X}}$, as opposed to $\sigma^2_X$ for the original signal) are sent through a channel. However, this may be a challenge for the visual system, since motion is encoded in spatial arrays of neurons rather than individual channels. A second use is actually carrying out the prediction to compensate for delays in perception or to drive motor output. (Note that since the position $q$ of a point is the integral of its velocity, the prediction of position by means of correlations in the velocity is given by $\hat{q}(t + \alpha) = q(t) + \int_0^\alpha \hat{v}(t + \tau)d\tau$, where $\hat{v}(t + \tau)$ is the prediction of the velocity at time $t + \tau$ given its history up to time $t$.) The results of this section indicate that this is feasible with or without taking the fluctuating scale into account.

## Discussion

The observed pattern of heavy-tailed velocity distributions in natural movies, with a scale parameter that is shared across velocity components and fluctuates in time, is remarkably consistent across scenes and categories, despite substantial variation in the content of those scenes, velocity correlation functions, and the overall velocity variance. Together with previous results showing similar statistics in natural images and sounds (*Schwartz and Simoncelli, 2001*), this suggests that scale-mixing is a fundamental property of natural stimuli, with deep implications for both neural coding and behavior.

In the context of object motion, scale mixing may arise from two distinct mechanisms, as outlined in our discussion of Brownian motion (see *Appendix 1*). First, objects may appear at a variety of distances from the observer, and those distances may change over time. The velocity of a point on an object, as it appears to an image-forming device, like a camera or the eye, is an angular velocity, which can be calculated as the tangential component of the physical velocity divided by the distance. A fluctuating distance thus scales the overall angular velocity over time: even an isolated point moving with Gaussian velocity statistics in three-dimensional space will have a heavy-tailed angular velocity distribution from the perspective of the observer. Second, the scale of the driving forces (either internal or external) may fluctuate over time. In our scenes, this corresponds to changes in the behavioral states of animals or to the turbulent nature of water and the wind driving plant motion. Since heavy-tailed distributions and scale fluctuations are observed in scenes with very little variance in depth, such as `bees8-full`, we emphasize that this mechanism is also at play in natural scenes.

Regardless of the source of scale-mixing, strategies for encoding and behaviorally compensating for it should be similar. On the encoding side, the presence of local scale variables suggests that sensory systems should adapt their response properties to local statistics in order to maximize information transmission. Given a fixed distribution of external input values, the optimal neural response function is the one that produces a uniform distribution over the neuron's dynamic range (*Laughlin,*

*1981*). The logarithmic speed tuning observed in MT (*Nover et al., 2005*) is consistent with this kind of static efficient coding. Here, we demonstrate that the scale of the distribution changes over time, so the gain of the response function should also change to match it (*Wainwright, 1999*; *Brenner et al., 2000*; *Wark et al., 2009*). Such adaptation or gain control is observed throughout the nervous system (see *Weber et al., 2019*, for a recent review), including in systems relevant to object motion encoding (*Fairhall et al., 2001*; *Bair and Movshon, 2004*; *Olveczky et al., 2007*; *Liu et al., 2016*). This adaptation could be the result of subcellular mechanisms, such as the molecular kinematics of synaptic vesicle release (*Ozuysal and Baccus, 2012*), or nonlinear circuit-level mechanisms (*Borst et al., 2005*; *Bharioke and Chklovskii, 2015*). By measuring the timescale on which the scale variable fluctuates in natural movie scenes, we have determined the timescale on which adaptation mechanisms in the brain should operate. Although the range is considerable, for most movies the time to 0.5 correlation for the scale generator is less than 1 s (*Figure 7I*). Future experiments could be targeted at probing adaptation timescales in the retina and cortex of various model organisms that occupy different environments. Our prediction is that these adaptation variables will tightly match the motion statistics in the organism's ecological niche.

Beyond single-cell adaptation, our results are also relevant to a population-level adaptation mechanisms known as divisive normalization (*Carandini and Heeger, 1994*; *Carandini and Heeger, 2012*), in which neighboring neurons in a population are mutually inhibitory in a divisive fashion. In many systems, motion is represented by a local population of neurons, each tuned to a narrow band of directions. Our results show that the fluctuating scale is shared between horizontal and vertical velocity components, and, hence, adaptation should ideally be distributed throughout the local population. Divisive normalization is a prime candidate for the implementation of this population-level adaptation, as has been suggested for GSM models of filter responses (*Schwartz and Simoncelli, 2001*; *Wainwright et al., 2002*; *Schwartz et al., 2006*; *Coen-Cagli et al., 2012*; *Coen-Cagli et al., 2015*; *Snow et al., 2016*). Most models of divisive normalization only capture steady-state responses to static or constant velocity stimuli, although some work has been done to describe the dynamics of divisive normalization during change detection and decision-making (*Louie et al., 2014*; *Ernst et al., 2021*). Again, these dynamics should be tuned to the timescale of the scale fluctuations measured here.

These data suggest a previously unexplored challenge for adaptation mechanisms in the context of object motion: an object may travel an appreciable distance before local mechanisms have a chance to take effect. A solution is to pool from a larger neighborhood, or, more intriguingly, for a local population to receive an adaptation signal selectively from those neurons in nearby populations whose preferred directions point to it. To our knowledge, these hypotheses have not yet been explored, either theoretically or experimentally.

In terms of behavior, our results help refine our understanding of the object tracking problems animals must solve in natural environments, which are crucial to survival. A commonly invoked framework for tracking is sequential Bayesian inference under a state-space model (*Ho and Lee, 1964*). In this framework, the brain has a probabilistic representation of the state of the object (i.e. a probability distribution over its position and velocity). An internal model of object motion is used to evolve this distribution forward in time, and this prediction is combined with incoming measurements to update the estimated state distribution. Under Gaussian assumptions, this yields the famous Kalman filter solution (*Kalman, 1960*). Our work has two important implications for the state-space model framework of object tracking. First, the velocity distributions we observe are typically non-Gaussian, so the Kalman filter solution is not strictly applicable. While heavy tails have little impact on prediction, they have a large effect on the uncertainty of the posterior estimate. Second, state-space models usually model the velocity as either an AR(1) or (discrete) diffusion process (i.e. a nonstationary AR(1) process with coefficient equal to one). The AR models we fit for the underlying Gaussian components generally have more than one large coefficient. The ARGSM model could naturally serve as a predictive state-space model that incorporates these empirical observations by including the recent history of the velocity and scale in the state description (note that the scale does not have a corresponding direct measurement, but it can be estimated from the incoming velocity measurements). Flexible Bayesian methods like the particle filter (*Del Moral, 1997*) can be used to implement such a model. The merging of the sort of adaptation mechanisms described above with neuromorphic particle filtering (*Kutschireiter et al., 2017*) is an intriguing avenue for future research.

Motion estimation itself can be framed as a Bayesian inference problem, and the tracking algorithm we use corresponds to a Gaussian prior (*Simoncelli et al., 1991*). The ARGSM model could thus serve as a better prior, motivating new motion estimation algorithms based on natural scene statistics. Speed perception in humans and animals can also be viewed through the lens of Bayesian inference, and experimental results are consistent with a heavy-tailed prior, specifically, a power law (*Stocker and Simoncelli, 2006*; *Zhang and Stocker, 2022*). The GSM model yields a heavy-tailed distribution for speed compared to the Rayleigh distribution expected under Gaussian assumptions, but it is not a true power law. Since power laws are an idealization and are always subject to some cutoff, the GSM model may be considered a more realistic (if less tractable) alternative. The correlated scale fluctuations also suggest that optimal Bayesian inference should be history-dependent, which could be assessed psychophysically using, e.g., a trial structure that is correlated in time.

Finally, the significant diversity of velocity and scale correlation functions and variances across scenes has implications both for efficient coding and for tracking. Namely, an encoder or tracker which is optimized for the statistics of one scene will be suboptimal for others. Indeed, there is a general trade-off in adaptation to global versus local statistics (*Młynarski and Hermundstad, 2018*; *Młynarski and Hermundstad, 2021*). The original efficient coding work posited adaptation on evolutionary timescales to natural scene statistics. Here, we emphasize the subsecond timescale of scale fluctuations in natural motion. Neural systems should also have the flexibility to adapt on intermediate timescales to changes in the environment or behavioral context (*Teşileanu et al., 2022*).

## Materials and methods
### Point tracking
We compute short trajectories using the `PointTracker` function in MATLAB's Computer Vision toolbox. The function employs a Kanade-Lucas-Tomasi (*Lucas and Kanade, 1981*; *Tomasi and Kanade, 1991*) feature tracking algorithm, which uses multi-scale image registration under a translational motion model to track individual points from frame to frame. Briefly, given an image patch $I(x, y, t)$ centered on some seeded initial position, the algorithm finds the displacement $(\Delta x, \Delta y)$ that minimizes the squared error,

$$\epsilon^2 = \iint \left[ I(x, y, t) - I(x + \Delta x, y + \Delta y, t + \Delta t) \right]^2 dxdy,$$

and updates the seed position on the next frame. Our strategy is to collect as many high-quality, short trajectories (64 frames) as possible from each movie, then subsample these down to a reasonable number of trajectories (8192) for statistical analysis. Initial points are seeded using the `detectMinEigenFeatures` function, which detects image features that can be tracked well under the motion model (*Shi and Tomasi, 1994*). From the initial seeds, we run the tracking algorithm forward and backward for 32 frames each, rather than running it in one direction for 64 frames. This increases the chances of capturing short-lived trajectories bounded by occlusion or image boundaries. Points are seeded on each frame, so the resulting set of trajectories is highly overlapping in time. On most movies, we employ the built-in forward-backward error checking method (*Kalal et al., 2010*), with a threshold of 0.25 pixels, to automatically detect tracking errors. The exceptions are three movies depicting water (water3, water5, and water6) where the small threshold leads to rejecting most trajectories, so we use a threshold of 8 pixels. In these cases, there are not well-defined objects, so relaxing this strict criterion is justified. The algorithm uses a multi-resolution pyramid and computes gradients within a neighborhood at each level. We use the default values of 3 pyramid levels and a neighborhood size of 31 by 31 pixels for all movies except the 3 water movies, where we find we can decrease the amount of erroneously large jumps in trajectories by increasing the neighborhood size to 129 by 129 pixels and using only 1 pyramid level (at a cost of greater computation time).

This method automatically tracks the stationary background points, which may be erroneously 'picked up' by a moving object as it traverses that location. To ensure that the trajectories we analyze are full of motion, we define a speed (velocity magnitude) threshold of 0.1 pix/frame and discard trajectories in which 16 or more time steps are below this threshold. We validated our method on a simple synthetic movie animated by an ARGSM process (*Appendix 2*).

The velocity time series is simply the first difference of the point positions along each trajectory. Within each ensemble, we subtract the ensemble mean from each velocity component (this is typically very close to zero, except for some water movies with a persistent flow). We then slightly rotate the horizontal and vertical velocity components to remove small correlations between them (these arise if, for example, objects tended to move along a slight diagonal relative to the camera's sensor). All visualizations and calculations are carried out after these minor preprocessing steps. Note that we do not subtract the average velocity within each trajectory, as this introduces an artificial anticorrelation at long lags.

## GSM models

The basic one-dimensional GSM model is described in the main text. Note that for some choices of the distribution for $S$, the distribution for $X$ has a closed-form solution. For example, the well-known Student's $t$-distribution is formed when $S$ follows an inverse $\chi$-distribution, and the Laplace distribution is formed when $S$ follows a Rayleigh distribution. In this work, we assume $S$ follows a log-normal distribution, which does not yield a closed-form distribution for $X$. This choice makes modeling correlations straightforward, as will be made clear below. In practice, the lack of a closed-form $p(x)$ is not a drawback, since we do not need to normalize the posterior distribution, $p(s|x) \propto p(x|s)p(s)$, in order to sample from it.

When considering multiple variables, a shared scale variable introduces a nonlinear form of dependence between them. Suppose $X_1 = Y_1 S$ and $X_2 = Y_2 S$. If $Y_1$ and $Y_2$ are uncorrelated, then $X_1$ and $X_2$ are conditionally independent given $S$:

$$p(x_1, x_2|s) = p(x_1|s)p(x_2|s).$$

However, $X_1$ and $X_2$ are not, in general, independent:

$$p(x_1, x_2) = \int_0^\infty p(x_1|s)p(x_2|s)p(s)ds \neq p(x_1)p(x_2).$$

This nonlinear dependence manifests itself in the elliptic level sets of $p(x_1, x_2)$, in contrast to the diamond-shaped level sets of $p(x_1)p(x_2)$. Note that this nonlinear dependence can coincide with the usual linear dependence if $Y_1$ and $Y_2$ are correlated, and that a weaker form of nonlinear dependence may be present if $X_1 = Y_1 S_1$ and $X_2 = Y_2 S_2$, where $S_1$ and $S_2$ are not independent.

## Autoregressive models

Autoregressive models (*Yule, 1927*; *Walker, 1931*) are a well-established and flexible way to capture correlations in time series data by supposing a linear relationship between the current value of a random variable with its previous values. Given a time series, $\{X_1, \ldots, X_T\}$, the $k$th order autoregressive, or AR($k$), model is given by

$$X_t = \sum_{i=1}^{k} \phi_i X_{t-i} + \xi_t$$

where $\{\phi_1, \ldots, \phi_k\}$ are regression coefficients and $\xi_t$ is Gaussian noise with variance $\sigma^2$.

The order $k$ is typically chosen by cross-validation to avoid overfitting. This makes sense from the standpoint of finding a model that generalizes well to new data. However, our primary aim here is simply to measure the autocovariances of the hidden variables, since their timescales are relevant to prediction and adaptation in the nervous system. For this reason, we choose $k$ to be as high as possible: $k = 31$ time steps, since $k$ must be less than $T/2$.

Typically, the model parameters are fit by standard linear regression (after organizing the data appropriately) (*Hamilton, 2020*). However, this method gives maximum likelihood estimates only if the initial $k$ time steps are considered fixed. If the initial data are assumed to be drawn from the stationary distribution defined by the parameters, the problem becomes nonlinear. The EM algorithm described below requires parameter estimates to be maximum likelihood, and since we would like the initial $k$ time steps (where $k$ is large) to be modeled by the stationary distribution, we must pursue this more difficult course. We calculate the maximum likelihood estimates numerically, following *Miller, 1995*. See *Appendix 3* for a full description of this method.

## The ARGSM model

The dynamic scale-mixture model generalizes the two-dimensional, shared scale variable GSM model described above to time series, assuming the underlying Gaussian random variables, $Y_1$ and $Y_2$, and the generator, $Z$, of the scale variable are all AR($k$) processes. Specifically, let $X_{1,t} = Y_{1,t}S_t$ and $X_{2,t} = Y_{2,t}S_t$. Written as $T$-dimensional vectors, we have $\mathbf{X}_1 = \mathbf{Y}_1 \odot \mathbf{S}$ and $\mathbf{X}_2 = \mathbf{Y}_2 \odot \mathbf{S}$, where $\odot$ is element-wise multiplication. The AR process assumptions imply

$$p(\mathbf{y}_1) = \mathcal{N}\left(\mathbf{y}_1; \mathbf{0}, \boldsymbol{\Sigma}_{Y_1}\right)$$
$$p(\mathbf{y}_2) = \mathcal{N}\left(\mathbf{y}_2; \mathbf{0}, \boldsymbol{\Sigma}_{Y_2}\right)$$
$$p(\mathbf{z}) = \mathcal{N}\left(\mathbf{z}; \mathbf{0}, \boldsymbol{\Sigma}_Z\right)$$

where the covariance matrices are determined by the parameters of independent AR($k$) models as described above. $\mathbf{S}$ is related to $\mathbf{Z}$ by element-wise application of the exponential function, $\mathbf{S} = \exp(\mathbf{Z})$. When $\mathbf{Z}$ is known, we have

$$p(\mathbf{x}_1|\mathbf{z}) = \mathcal{N}\left(\mathbf{x}_1; \mathbf{0}, \mathbf{D_s}\boldsymbol{\Sigma}_{Y_1}\mathbf{D_s}\right)$$
$$p(\mathbf{x}_2|\mathbf{z}) = \mathcal{N}\left(\mathbf{x}_2; \mathbf{0}, \mathbf{D_s}\boldsymbol{\Sigma}_{Y_2}\mathbf{D_s}\right)$$

where $\mathbf{D_s}$ is the matrix with the elements of $\mathbf{s}$ along the diagonal and zeros elsewhere.

## EM and stochastic approximation

The classic EM algorithm is a useful tool for finding (local) maximum likelihood estimates of parameters of hidden variable models (*Dempster et al., 1977*). Let $\theta = \left\{\phi_{Y_1}, \phi_{Y_2}, \phi_Z, \sigma^2_{Y_1}, \sigma^2_{Y_2}, \sigma^2_Z\right\}$ be the collection of parameters of the ARGSM model, where each $\phi$ is the vector of AR coefficients and each $\sigma^2$ is the driving noise variance for each variable. The observed data, $\mathcal{D} = \{\mathbf{x}_{1,n}, \mathbf{x}_{2,n}\}, 1 \le n \le N$, are the $N$ pairs of $T$-dimensional vectors corresponding here to the horizontal and vertical velocity along each trajectory. The hidden variables, $\mathcal{H} = \{\mathbf{z}_n\}, 1 \le n \le N$, are the Gaussian generators of the time-varying scale associated with each trajectory. The likelihood

$$
\begin{aligned}
L &= \log p(\mathcal{D}|\theta) \\
&= \sum_{n=1}^{N} \log \int_{\mathbb{R}^T} p(\mathbf{x}_{1,n}, \mathbf{x}_{2,n}|\mathbf{z}, \theta)p(\mathbf{z}|\theta)D\mathbf{z}
\end{aligned}
$$

is intractable to maximize due to the high-dimensional integral. The EM algorithm finds a local maximum iteratively. Starting with an initial guess for the parameters, $\theta_0$, at each step, one computes the expectation with respect to the probability distribution of the hidden variables given the data and the current parameter estimate $\theta_t$ of the complete data log-likelihood,

$$Q(\theta|\theta_t) = E_{p(\mathcal{H}|\mathcal{D},\theta_t)}\left[\log p(\mathcal{D}, \mathcal{H}|\theta)\right],$$

then updates the parameters to maximize this function,

$$\theta_{t+1} = \arg\max_{\theta} Q(\theta|\theta_t).$$

In our setting, we have

$$
\begin{aligned}
Q(\theta|\theta_t) &= -\frac{1}{2}\sum_{n=1}^{N} E\left[\mathbf{y}_{1,n}^{\top}\boldsymbol{\Sigma}_{Y_1}^{-1}\mathbf{y}_{1,n} + \mathbf{y}_{2,n}^{\top}\boldsymbol{\Sigma}_{Y_2}^{-1}\mathbf{y}_{2,n} + \mathbf{z}_n^{\top}\boldsymbol{\Sigma}_Z^{-1}\mathbf{z}_n\right] + K \\
&= -\frac{1}{2}\mathrm{tr}\left(\mathbf{C}_{Y_1}\boldsymbol{\Sigma}_{Y_1}^{-1} + \mathbf{C}_{Y_2}\boldsymbol{\Sigma}_{Y_2}^{-1} + \mathbf{C}_Z\boldsymbol{\Sigma}_Z^{-1}\right) + K,
\end{aligned}
$$

where

$$\mathbf{C}_Z = \sum_{n=1}^{N} E\left[\mathbf{z}_n\mathbf{z}_n^{\top}\right]$$

and similarly for $\mathbf{C}_{Y_1}$ and $\mathbf{C}_{Y_2}$. Note that in this context, the $\mathbf{y}$ variables are merely shorthand for $\mathbf{y}_{1,n} = \mathbf{x}_{1,n} \oslash \mathbf{s}_n$ and $\mathbf{y}_{2,n} = \mathbf{x}_{2,n} \oslash \mathbf{s}_n$, where $\oslash$ is element-wise division. Given the $\mathbf{C}$ matrices, the maximum likelihood estimates of the AR parameters can be calculated as described in *Appendix 3*.

Unfortunately, the expectation values in the $\mathbf{C}$ matrices are also intractable, but they can be approximated through sampling methods. A variant of the EM algorithm, called stochastic approximation EM, was developed to address this problem (**Delyon et al., 1999**). Given a sample from the distribution $p(\mathcal{H}|\mathcal{D}, \theta_t)$, one calculates the sample matrices $\hat{\mathbf{C}}$, then updates the stochastic approximations as

$$\mathbf{C}_t = \mathbf{C}_{t-1} + \eta_t \left( \hat{\mathbf{C}}_t - \mathbf{C}_{t-1} \right) .$$

The sequence of parameters $\eta_t$ is given by

$$\eta_t = \begin{cases} 1 & 1 \leq t \leq \alpha \\ (t - \alpha)^{-\beta} & t > \alpha \end{cases} .$$

We choose $\alpha = 2500$ or $5000$, so that the algorithm runs in a fully stochastic mode until the parameter estimates are nearly stationary, and $\beta = 1$, so that after this initial period, the algorithm converges by simply taking a running average of the samples of the $\hat{\mathbf{C}}$ matrices. Importantly, the samples do not need to be independent across iterations for the algorithm to converge (**Kuhn and Lavielle, 2004**). This means that, when performing the Gibbs sampling described below, we only need to update each hidden variable element once for each iteration, rather than updating many times and throwing out samples to achieve independence. Since each M-step (the AR model MLE algorithm described above) is much faster than each E-step (calculating the $\mathbf{C}$ matrices through sampling), this results in a more sample-efficient algorithm (**Neal and Hinton, 1998**).

We also estimate the expectation of the hidden variables $\{\mathbf{z}_n\}$ in an identical fashion. This is equivalent to a Bayesian point estimate where the estimated parameters form a forward model and prior. These estimates are then used to remove the scale from the velocity, in order to examine the kurtosis under different model assumptions (**Figure 4**).

The EM algorithm, and its stochastic approximation variant, converges to a local maximum of the likelihood function that depends on the initial conditions. We find that, in practice, it is important to introduce the scale variable gradually to the model. We initialize the model with AR parameters fit to the raw data for the $Y_1$ and $Y_2$ components, and let $Z$ be uncorrelated with very small variance (regression coefficients $\phi_Z = \mathbf{0}$ and driving noise variance $\sigma_Z^2 = 0.05^2(\gamma_{Y_1,0} + \gamma_{Y_2,0})/2$).

## Sampling methods

We use a combination of Gibbs and rejection sampling to sample from the posterior of the hidden variables given the data and the current parameter estimates (**Bishop, 2006**). In Gibbs sampling, an initial vector $\mathbf{z}$ is used to generate a new sample by sampling each element individually, conditioned on the remaining elements. Since the conditional distribution is intractable, we use rejection sampling, which allows us to sample from an arbitrary, unnormalized distribution by sampling from a proposal distribution (in this case a Gaussian with parameters chosen to envelope the conditional distribution) and rejecting some draws in order to shape it into the target distribution. See *Appendix 3* for a detailed description of the sampling algorithm.

## Acknowledgements

This work was supported by the National Science Foundation through the Physics Frontier Center for Living Systems (PHY-2317138), the Center for the Physics of Biological Function (PHY-1734030), and a CAREER award to SEP (IIS-1652617); by the NSF-Simons National Institute for Theory and Mathematics in Biology, awards DMS-2235451 (NSF) and MP-TMPS-00005320 (Simons Foundation); and by the National Institutes of Health BRAIN Initiative (R01EB026943). We thank Siwei Wang and Benjamin Hoshal for useful comments on the manuscript.

## Additional information

### Funding

| Funder | Grant reference number | Author |
|--------|------------------------|--------|
| National Science Foundation | PHY-2317138 | Stephanie E Palmer |
| National Science Foundation | PHY-1734030 | Stephanie E Palmer |
| National Science Foundation | IIS-1652617 | Stephanie E Palmer |
| National Science Foundation | DMS-2235451 | Stephanie E Palmer |
| Simons Foundation | MP-TMPS-00005320 | Stephanie E Palmer |
| National Institutes of Health | R01EB026943 | Stephanie E Palmer |

The funders had no role in study design, data collection and interpretation, or the decision to submit the work for publication.

### Author contributions

Jared M Salisbury, Conceptualization, Data curation, Software, Formal analysis, Validation, Investigation, Visualization, Methodology, Writing – original draft, Writing – review and editing; Stephanie E Palmer, Conceptualization, Data curation, Supervision, Funding acquisition, Investigation, Methodology, Writing – original draft, Project administration, Writing – review and editing

### Author ORCIDs

Jared M Salisbury ⓘD https://orcid.org/0009-0004-0215-9717
Stephanie E Palmer ⓘD https://orcid.org/0000-0001-6211-6293

### Decision letter and Author response

Decision letter https://doi.org/10.7554/eLife.104054.sa1
Author response https://doi.org/10.7554/eLife.104054.sa2

## Additional files

### Supplementary files

MDAR checklist

### Data availability

All videos analyzed are part of the Chicago Motion Database, located at https://cmd.rcc.uchicago.edu. Code for fitting the ARGSM model is available at https://github.com/sepalmer/motion-scale-mixture (copy archived at *Palmer and Salisbury, 2026*).

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

# Appendix 1

## A motivating example: Brownian motion

Here, we introduce a canonical example of natural motion to guide our intuition about departures from the simplest case: Brownian motion. Brown discovered the motion that bears his name by observing small particles released by a grain of pollen floating in water under a microscope (**Brown, 1828**) – a highly controlled setting, but similar in spirit to our own. Each particle is subjected to tiny molecular forces from individual water molecules colliding with it at random. As a result, it moves in a random fashion across the surface of the water, limited only by the boundary of the dish. Its motion is not completely unpredictable, however; since the particle has some mass, albeit small, it has some inertia, or tendency to continue moving with the same velocity. Statistically, this means the velocity is correlated in time.

Based on this observation, Einstein derived his famous diffusion equation, describing how a density of diffusing particles changes over time, by considering the case of particles with infinitesimal mass (**Einstein, 1905**). For describing the motion of an individual, massive particle, it is more useful to look to Langevin's description, an early application of stochastic differential equations (**Langevin, 1908**). Using Newton's law and assuming the force can be divided into the sum of a drag force, proportional to the velocity by a factor of $-\gamma$, and a randomly fluctuating force due to the collision of water molecules, $F(t)$, we have the following differential equation for the velocity, $V(t)$, of a particle with mass $m$:

$$m\frac{d}{dt}V(t) = -\gamma V(t) + F(t).$$

When $F(t)$ is an uncorrelated Gaussian process, $V(t)$ is an Ornstein-Uhlenbeck process, which has an exponential correlation function (see **Gillespie, 1996a**; **Gillespie, 1996b** for a pedagogical review). The variance of the velocity, $\sigma^2$, is related to the temperature, $T$, by $\sigma^2 = kT/m$, where $k$ is Boltzmann's constant.

Numerically and experimentally, we must always discretize time. The discrete-time approximation to the Ornstein-Uhlenbeck process with time-step $\Delta t$ is given by the difference equation

$$V_{t+\Delta t} = \phi V_t + \xi_t,$$

$$p(\xi_t) = \mathcal{N}\left(\xi_t; 0, \sigma_\xi^2\right),$$

which is a first-order autoregressive, or AR(1), process (**Yule, 1927**; **Walker, 1931**). Like the Ornstein-Uhlenbeck process, it has an exponential autocovariance function, given by

$$E\left[V_t V_{t+k\Delta t}\right] = \sigma^2 \exp\left(-\frac{k\Delta t}{\tau}\right),$$

with variance

$$\sigma^2 = \frac{\sigma_\xi^2}{1 - \phi^2}$$

and time constant

$$\tau = -\frac{\Delta t}{\ln \phi}.$$

Let us consider how to incorporate a fluctuating variance in the AR(1) process. There are two possibilities: the variance of the driving noise $\xi_t$ could fluctuate, as in Brownian motion with a fluctuating temperature. Alternatively, the entire process could be scaled by a fluctuating positive variable; in the Brownian motion experiment, this corresponds to changes in the magnification level of the microscope. This also corresponds to changes in inverse distance if the velocity is an angular velocity relative to an observer. The former, additive model corresponds to

$$V^+_{t+\Delta t} = \phi V^+_t + S_t \xi_t.$$

The latter, multiplicative model corresponds to

$$V^\times_t = S_t V_t,$$

where $S_t$ is the fluctuating scale. Rewriting

$$V^\times_{t+\Delta t} = \frac{S_{t+\Delta t}}{S_t} \phi V^\times_t + S_{t+\Delta t} \xi_t,$$

we see that $V^+_t$ and $V^\times_t$ are identical when $S_t$ is constant in time and varies only across the ensemble. However, they are not equivalent when $S_t$ changes over time. In the main text, we adopt the multiplicative model, since it factors into a product of two independent stochastic processes (one for normalized velocity and one for scale), which simplifies the inference problem.

## Appendix 2

### Tracking algorithm details

Validation

We validated the tracking algorithm by generating a synthetic movie, computing trajectories, and comparing them to ground truth. The synthetic movie consists of a single frame from `bees8-full` with a grid of 16 disks (of radius 48 pixels) copied from their initial positions, overlaid on the original image, and animated according to an ARGSM process. The underlying Gaussian velocity time series were generated by simulating critically damped harmonic oscillators (independently for each component) with velocity standard deviation $\sigma_{Y_1} = \sigma_{Y_2} = 0.5$ and time constant $\tau = 100$ ms. The scale time series were generated by simulating an AR(1) process with standard deviation $\sigma_Z = 0.5$ and coefficient $\phi_Z = 0.75$. We multiplied each pair of velocity components by the scale and calculated the cumulative sum to determine the position of the disks on each frame relative to their grid positions. The disks occasionally occlude one another but are largely isolated; we expect that adding more disks and occlusions decreases tracking performance accordingly.

We apply the tracking algorithm to generate 8192 trajectories of length 65. After applying the tracking algorithm, we associate each tracking trajectory with a given ground truth trajectory if its initial position is within a given radius of the center of the corresponding disk; trajectories which are not associated with any ground truth are discarded. We calculate the mean squared error of the velocity along these trajectories relative to the ground truth velocity at the corresponding times. Looking at the fraction that remains (*Appendix 2—figure 1A*), we find that many trajectories are seeded at the perimeter of the disk, since there tends to be a distinct edge there. The presence of trajectories with initial positions outside of any disk indicates that the algorithm detects some spurious motion in the vicinity of the actual motion. We calculate the mean squared error of the velocity along these trajectories relative to the ground truth velocity at the corresponding times. The SNR (*Appendix 2—figure 1B*), or ratio of the variance of the ground truth velocity to the mean squared error, drops off quickly when the radius is large enough to include the spurious trajectories. However, the statistics of the ensemble are very close to the ground truth when all of the tracking trajectories are included. The distribution of a single velocity component (*Appendix 2—figure 1C*) captures the overall shape and tails of the ground truth distribution but is more sharply peaked. The autocorrelation (*Appendix 2—figure 1D*) is underestimated only slightly.

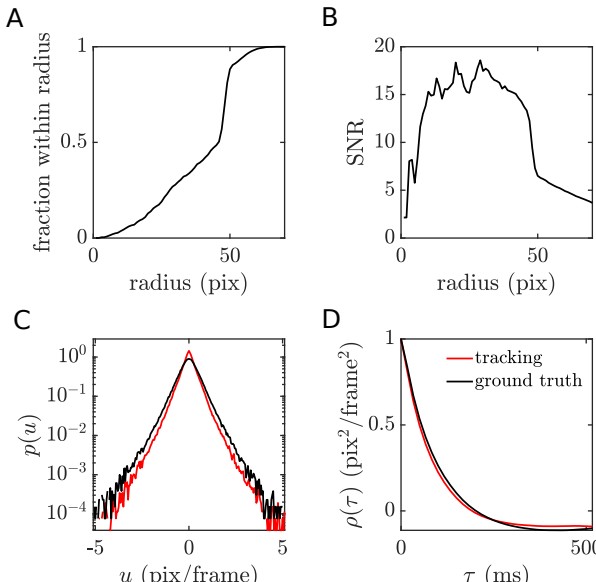

**Appendix 2—figure 1.** Tracking performance. (**A**) Fraction of trajectories within a given radius of the center of the moving disks. (**B**) Tracking signal-to-noise ratio for the subset of trajectories within a given radius. (**C**) Horizontal velocity histogram for the tracking data and ground truth. (**D**) Horizontal velocity autocorrelations for the tracking data and ground truth.

## Effect of restricting velocity to trajectories

Since tracking is a difficult problem that involves associating velocity values in time subject to our criteria for trajectory quality, the set of velocity values along trajectories is only a subset of all velocity values in a given scene. To understand the difference, we apply the same algorithm with the trajectory length set to two, so there is only a single velocity per trajectory, with no forward-backward error thresholding. For `bees8-full`, we find a substantially broader distribution for the unrestricted velocity (see **Appendix 2—figure 2**). The sample standard deviation and kurtosis for the trajectory-restricted velocity are 1.2 pix$^2$ and 10.2, respectively, compared to 1.9 pix$^2$ and 24.5 for the unrestricted velocity. We thus expect the standard deviation and kurtosis values reported here to be underestimates compared to theoretically perfect tracking.

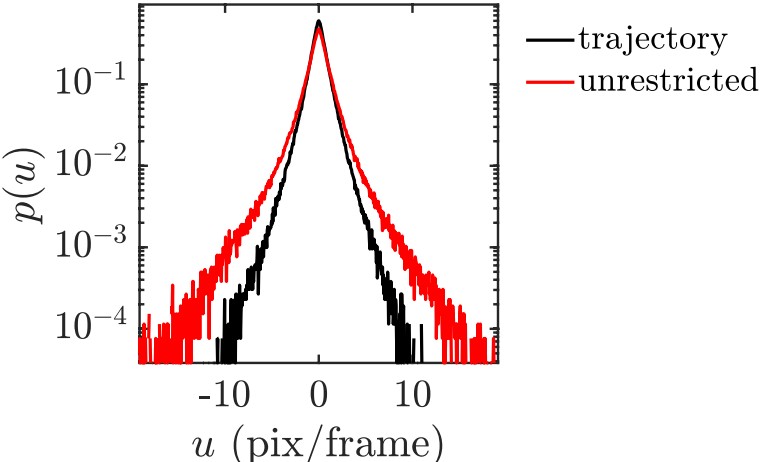

**Appendix 2—figure 2.** Velocity histograms. Histograms for horizontal velocity with and without restriction to trajectories.

## Appendix 3

### Numerical details

#### Maximum likelihood estimation for autoregressive models

Following *Miller, 1995*, we consider the time series as a $T$-dimensional vector, $\mathbf{X} = [X_1 \ldots X_T]^\top$. Then,

$$p(\mathbf{x}) = \mathcal{N}\left(\mathbf{x}; \mathbf{0}, \boldsymbol{\Sigma}\right),$$

where

$$\boldsymbol{\Sigma} = E\left[\mathbf{X}\mathbf{X}^\top\right] = \text{Toeplitz}(\gamma_0, \ldots, \gamma_{T-1}).$$

The autocovariance series $\{\gamma_0, \ldots, \gamma_{T-1}\}$ is related to the regression coefficients and driving noise variance by

$$\gamma_0 = \sum_{i=1}^{k} \phi_i \gamma_i + \sigma^2$$

$$\gamma_j = \sum_{i=1}^{k} \phi_i \gamma_{i-j} \qquad j \geq 1.$$

Given a collection of $N$ samples of length $T$ from the time series, we calculate the matrix $\mathbf{R}$ defined by

$$R_{ij} = R_{ji} = \frac{1}{N} \sum_{n=1}^{N} \sum_{t=i+1}^{T-j} x_{n,t} x_{n,t+i-j} \qquad 0 \leq i \leq j \leq k,$$

where the index $n$ ranges over samples and $t$ ranges over time steps. Then, the maximum likelihood estimate of the parameters satisfies a nonlinear system of $2(k+1)$ equations, consisting of the $k+1$ autocovariance relations above, up to $\gamma_k$, together with the following:

$$R_{00} = \sum_{i=1}^{k} \phi_i \left(R_{0i} + i\gamma_i\right) + T\sigma^2$$

$$R_{j0} = \sum_{i=1}^{k} \phi_i \left(R_{ij} + i\gamma_{i-j}\right) \qquad 1 \leq j \leq k.$$

We solve this system numerically using MATLAB's `fsolve`.

While the covariance matrix is easily calculated from the AR model parameters, it is not well appreciated that the inverse covariance matrix can be calculated exactly from the parameters as well (*Verbyla, 1985*). This is useful because it is the inverse that is needed for the sampling methods described below, and inverting large, nearly singular matrices is numerically unstable. The inverse of $\boldsymbol{\Sigma}$ is given by

$$\boldsymbol{\Sigma}^{-1} = \frac{1}{\sigma^2} \left( \mathbf{I} + \sum_{i=1}^{k} \phi_i^2 \mathbf{E}_i - \sum_{i=1}^{k} \phi_i \mathbf{F}_i + \sum_{i=1}^{k-1} \sum_{j=1}^{p-i} \phi_i \phi_{i+j} \mathbf{G}_{j,i+j} \right),$$

where $\mathbf{I}$ is the identity matrix, $\mathbf{E}_i$ is the identity matrix with the first and last $i$ diagonal elements set to zero, $\mathbf{F}_i$ is a matrix with ones along the $i$th upper and lower diagonals and zeros elsewhere, and $\mathbf{G}_{j,i+j} = \mathbf{E}_i \mathbf{F}_j \mathbf{E}_i$ (note that we have corrected a typo in *Verbyla, 1985*, in the indexing of the last term of the equation).

## Sampling methods

Gibbs sampling works by starting with an initial vector $\mathbf{z}$ and updating each element $z_t$ (in a random order) conditioned on the remaining elements $\mathbf{z}_{\backslash t}$, where the notation $\backslash t$ indicates all indices in the set $\{1, \ldots, T\} \backslash t$. The posterior distribution is given by

$$
\begin{aligned}
p(z_t | \mathbf{x}_1, \mathbf{x}_2, \mathbf{z}_{\backslash t}) \quad & \propto p(x_{1,t}, x_{2,t} | \mathbf{x}_{1,\backslash t}, \mathbf{x}_{2,\backslash t}, \mathbf{z}) p(z_t | \mathbf{x}_{1,\backslash t}, \mathbf{x}_{2,\backslash t}, \mathbf{z}_{\backslash t}) \\
& = p(x_{1,t} | \mathbf{x}_{1,\backslash t}, \mathbf{z}) p(x_{2,t} | \mathbf{x}_{2,\backslash t}, \mathbf{z}) p(z_t | \mathbf{z}_{\backslash t}) \,.
\end{aligned}
$$

Each distribution in the product is a Gaussian given by

$$
p(x_{1,t} | \mathbf{x}_{1,\backslash t}, \mathbf{z}) = \mathcal{N} \left( x_{1,t}, \hat{\mu}_{Y_1} s_t, \hat{\sigma}_{Y_1}^2 s_t^2 \right)
$$

$$
p(x_{2,t} | \mathbf{x}_{2,\backslash t}, \mathbf{z}) = \mathcal{N} \left( x_{2,t}, \hat{\mu}_{Y_2} s_t, \hat{\sigma}_{Y_2}^2 s_t^2 \right)
$$

$$
p(z_t | \mathbf{z}_{\backslash t}) = \mathcal{N} \left( \hat{\mu}_Z, \hat{\sigma}_Z^2 \right) \,,
$$

where

$$
\hat{\sigma}_{Y_1}^2 = \left( \boldsymbol{\Sigma}_{Y_1}^{-1} \right)_{tt}^{-1}
$$

$$
\hat{\sigma}_{Y_2}^2 = \left( \boldsymbol{\Sigma}_{Y_2}^{-1} \right)_{tt}^{-1}
$$

$$
\hat{\sigma}_Z^2 = \left( \boldsymbol{\Sigma}_Z^{-1} \right)_{tt}^{-1}
$$

$$
\hat{\mu}_{Y_1} = -\hat{\sigma}_{Y_1}^2 \left( \boldsymbol{\Sigma}_{Y_1}^{-1} \right)_{t,\backslash t} \mathbf{y}_{1,\backslash t}
$$

$$
\hat{\mu}_{Y_2} = -\hat{\sigma}_{Y_2}^2 \left( \boldsymbol{\Sigma}_{Y_2}^{-1} \right)_{t,\backslash t} \mathbf{y}_{2,\backslash t}
$$

$$
\hat{\mu}_Z = -\hat{\sigma}_Z^2 \left( \boldsymbol{\Sigma}_Z^{-1} \right)_{t,\backslash t} \mathbf{z}_{\backslash t} \,.
$$

Rejection sampling is used to draw a sample of each individual $z_t$. In this framework, a proposal distribution that is easy to sample from, $q(z)$, is chosen as an envelope (after an appropriate scaling) of a target distribution, $\hat{p}(z)$, from which we would like to sample, i.e., $\hat{p}(z) \leq M q(z)$ for some positive scaling factor $M$. The proposal distribution is sampled from, followed by sampling another variable $u$ from a uniform distribution. If $u \leq M q(z) / \hat{p}(z)$, the sample is accepted; otherwise, it is rejected and the sampling is repeated. The method produces samples from the target distribution exactly, even if it is unnormalized, but requires the envelope to be tight to avoid rejecting too many samples. Here, the target distribution is the unnormalized posterior (dropping time indices for clarity),

$$
\hat{p}(z) = \mathcal{N} \left( x_1; \hat{\mu}_{Y_1} s, \hat{\sigma}_{Y_1}^2 s^2 \right) \mathcal{N} \left( x_1; \hat{\mu}_{Y_2} s, \hat{\sigma}_{Y_2}^2 s^2 \right) \mathcal{N} \left( z; \hat{\mu}_Z, \hat{\sigma}_Z^2 \right) \,.
$$

We let the proposal distribution be a Gaussian,

$$
q(z) = \mathcal{N} \left( z; \mu_q, \sigma_q^2 \right) \,,
$$

whose mean and variance are optimized heuristically to form a tight envelope of the target distribution. We center this Gaussian over the peak of the target distribution,

$$
\mu_q = \arg \max_z \hat{p}(z) \,.
$$

and compute the scale factor $M$ to match the two distributions at the peak, $M = \hat{p}(\mu_q) / q(\mu_q)$. To optimize the variance, note that we can write

$$\log \hat{p}(z) = \quad -\frac{1}{2}\frac{\left[x_1 \exp(-z) - \hat{\mu}_{Y_1}\right]^2}{\hat{\sigma}_{Y_1}^2} - \frac{1}{2}\frac{\left[x_2 \exp(-z) - \hat{\mu}_{Y_2}\right]^2}{\hat{\sigma}_{Y_2}^2}$$
$$-\frac{1}{2}\frac{\left[z - \left(\hat{\mu}_Z - 2\hat{\sigma}_Z^2\right)\right]^2}{\hat{\sigma}_Z^2} - 2\left(\hat{\mu}_Z - \hat{\sigma}_Z^2\right) + K.$$

The first two terms rapidly approach constants to the right and diverge negatively to the left. Thus, the right tail of $\hat{p}(z)$ behaves as $\mathcal{N}(z; \hat{\mu}_Z - 2\hat{\sigma}_Z^2, \hat{\sigma}_Z^2)$, and the left tail falls off extremely rapidly. A variance of $\sigma_q^2 = \hat{\sigma}_Z^2$ is necessary and sufficient to cover the two tails when the peak of the proposal distribution is to the right of the peak of this Gaussian distribution, i.e., when $\mu_q \geq \hat{\mu}_Z - 2\hat{\sigma}_Z^2$. A smaller variance will not fully envelop the right tail regardless of the value of $\mu_q$. We require a slightly larger variance when $\mu_q < \hat{\mu}_Z - 2\hat{\sigma}_Z^2$, which we calculate by assuming $Mq(z)$ makes exactly one other point of contact with $\hat{p}(z)$. Let

$$f(z) = \log \hat{p}(z)$$

and

$$g(z) \quad = \log[Mq(z)]$$
$$= -\frac{1}{2}\frac{(z - \mu_q)^2}{\sigma_q^2} + f(\mu_q).$$

Let $z_0$ be the point of contact. Then, $f(z_0) = g(z_0)$ implies

$$\sigma_q^2 = \frac{1}{2}\frac{(z_0 - \mu_q)^2}{f(\mu_q) - f(z_0)}.$$

The two curves must also be tangent at $z_0$, $f'(z_0) = g'(z_0)$. We have

$$g'(z_0) = -\frac{z_0 - \mu_q}{\sigma_q^2} = -2\frac{f(\mu_q) - f(z_0)}{z_0 - \mu_q}.$$

The tangency condition implies $z_0$ is the solution to

$$f'(z_0) + 2\frac{f(\mu_q) - f(z_0)}{z_0 - \mu_q} = 0,$$

which we solve numerically using MATLAB's `fzero` function, and use this solution to calculate $\sigma_q^2$. Finally, if the above procedure fails due to numerical issues, we simply set $\sigma_q^2 = (1.1\hat{\sigma}_z)^2$, which is large enough to form an envelope in practice.

