## [Editor Report]

This paper tackles an important problem: the statistics of natural motion. The statistics of natural stimuli are in general highly structured, and the properties of that structure has guided understanding of sensory coding. This paper extends this analysis to natural motion. The authors first characterize the non-Gaussian properties of natural motion, and then introduce a simple Gaussian scale-mixture model that captures that behavior. The model is developed in a clear and convincing way and the results are compelling.

---

## [Decision Letter]

**Decision letter after peer review:**

Thank you for submitting your article "A dynamic scale-mixture model of motion in natural scenes" for consideration by *eLife*. Your article has been reviewed by 3 peer reviewers, and the evaluation has been overseen by a Reviewing Editor and Timothy Behrens as the Senior Editor.

The reviewers all appreciated the importance of the topic and the general approach taken. The main concerns regarded the clarity of explanations of several key results and assumptions for some of the analyses. Those are detailed in the reviewer comments below.

*Reviewer #1 (Recommendations for the authors):*

Strengths:

A systematic treatment of the properties of natural motion is needed. The point tracking approach used here is a nice approach to resolve many of the challenges facing other approaches to this problem. The evidence that motion statistics are non-Gaussian is clear, and the introduction of the nonlinear scaling parameter to capture that non-Gaussian behavior provides a quite useful characterization of that non-Gaussian behavior.

Weaknesses:

1. Intuitively, it seems expected that horizontal and vertical velocities should be correlated given that most motion will occur in a direction that is not purely along one of those axes. That intuitive description seems to capture the key features illustrated in Figure 2. Can this physical aspect of the problem be incorporated into the analysis? At a minimum more could be done to develop this intuition, and then use that to intuition to help interpret the more quantitative analyses.

2. Some of the assumptions in the "Coding implications of heavy tails" section were not made sufficiently clear. First, I think that a nonlinearity that mapped the distribution to a Gaussian would introduce signal dependent noise only if the noise occurs prior to the nonlinearity. If that is correct, it should be stated as part of that analysis. Second, doesn't the "demodulation" approach introduced on line 219 also introduce signal dependent noise if the noise is introduced prior to the normalization step? This would seem to violate the conditions under which the simple Gaussian channel argument holds (line 199-201). Because of these concerns it was not clear what to take away from this section.

3. Figure 5A and B are described (first paragraph of the dynamics section) as providing evidence for a contribution of a nonlinear scale factor to motion correlations. I am not clear on why that is needed. Both of these would seem to be true for simple linear correlations. I may well be missing something, but if so the argument in the text about the need for the nonlinear scaling in this case should be made more clearly. This concern made it difficult to evaluate the rest of that section, as it was not clear why the scaling was needed. This figure and the associated analysis could be described more clearly, perhaps showing the bow-tie structures from earlier figures.

Line 206: I believe the reference here should be to Figure 4G

*Reviewer #2 (Recommendations for the authors):*

Summary of work:

The paper introduces a model of motion in video sequences from natural scenes. The main contribution of the paper is to show that the statistics of the velocity of the motion of key points in natural scenes are well captured by Gaussian scale mixtures (GSM) models. Two hypotheses for this are that: (1) the distance of the moving objects to the camera control the range of angular motion; and (2) the scale of the driving forces that create motion may fluctuate over time. The proposed model introduces a GSM process which is a combination of an underlying Gaussian process and a scaling process.

Strengths:

- The proposed model elegantly captures the statistics of the velocity distribution for vertical and horizontal direction in the video sequence.

- The authors provide an intuitive information theoretic justification for the normalization process that involves reversing the scaling operation to recover the underlying Gaussian latent factor of their model.

- The justification of the model's complexity and the gains in predictive accuracy using the Akaike Information Criterion (AIC) add strength to the argument.

- The choice of the distribution of the scaling variable as a $\log$ normal random variable provides an easy way to track model in terms of relevant statistics like variance and kurtosis that apply to testing fits and justifying the coding efficiency arguments.

- The introduction of a dynamic GSM using Gaussian processes for both the underlying Gaussian variable and the $\log$ of the scaling factors is also well justified with comparison to other alternative models with static scaling or time independence.

Weaknesses:

- While the model using $\log$ normal scaling variables provides tractable statistics it may not be the most accurate and also not fully justified from the underlying phenomenon (at least this point is not made clear in the paper)

- The proposed model discusses a way to represent velocities as a multiplicative mix. It would benefit from making more explicit connection between the GSM model's representation and areas in the brain that may represent velocities. If the work in this area is lacking, does the proposed model hint at areas that could be encoding velocities?

*Reviewer #3 (Recommendations for the authors):*

This is one of the most enjoyable manuscripts I have reviewed in a while! The topic of object motion estimation and tracking is central to visual and visuomotor processing. While this is a very broad topic, the manuscript focuses on a small, well-defined part of it and reports a study with exemplary execution. The main finding is that the statistics of motion of points (a more tractable proxy for "objects") in natural videos are heavy-tailed, highly predictable, and well modeled with Gaussian Scale Mixtures and their new dynamic (auto-regressive) extensions. The main implications are that these regularities should be exploited by the visual system to predict object motion and by neurons to adapt to object motion statistics across different environments, leading to several experimental predictions.

The authors propose to characterize the statistics of local motion over time in natural videos, and to develop generative models that capture those statistics. To do so, they choose a database of natural videos with moderate complexity, including no camera motion and closeup scenes of uninterrupted dense motion (e.g. water flowing or swarms of insects). This allows them to use effectively a standard point-tracking algorithm to define the point motion trajectories for analysis, and to restrict their analysis to 1-second-long clips of uninterrupted but substantially variable motion. They explain well the importance of studying trajectories (which are dynamic motion features) rather than simply motion energy or optic flow, which are in a sense instantaneous motion features.

Their analysis provides compelling evidence that both the instantaneous and dynamic statistic of point velocities deviate from Gaussian, have long tails, and may result from a Gaussian-distributed velocities multiplied by a global scaling factor, leading them to formulate Gaussian Scale Mixture models (GSM). GSMs have been used to model statistics of images (some references cited) and instantaneous statistics of movies (some references missing), but not in the context of object/point motion and not considering entire trajectories, as they do here. The analysis carefully quantifies the improvement in capturing motion statistics with their models over simpler alternatives.

[Editors' note: further revisions were suggested prior to acceptance, as described below.]

Thank you for resubmitting your work entitled "A dynamic scale-mixture model of motion in natural scenes" for further consideration by *eLife*. Your revised article has been evaluated by Timothy Behrens (Senior Editor) and a Reviewing Editor.

The manuscript has been improved but there are some small remaining suggestions that should be considered, as detailed below:

*Reviewer #1 (Recommendations for the authors):*

This is a revision of a paper describing the statistics of natural motion. This is an issue that, to my knowledge, has not been described in detail and the work in the paper fills this gap nicely. The revisions have improved the paper substantially. I have a few remaining suggestions.

Line 117: define "good trajectories"

Line 130-132: the heavy tails are clear in Figure 2 but not in Figure 1. Can you refer ahead here to help a reader out?

Line 207 (and below): doesn't this strategy often also increase noise since it must be based on an estimate of X (with noise) and the estimate of S could be small in some instances?

Line 286: define "innovation noise" – particularly why it is given that name

Line 287: why is the innovation noise small?

Lines 291-295: What is going on with the insect movies?

Lines 291-295: How can the variance explained by the naive model R^2_X be higher than that for the more complete model?

Lines 302-304: the prediction section ends on a technical note. It would be helpful to have a summary of the main take home point from that section.

*Reviewer #2 (Recommendations for the authors):*

After reading the revised version of the manuscript, I can see that the concerns raised by all reviewers where either incorporated into the manuscript or clarified in the author responses. The section on the coding implications of heavy tails presents the argument better as it also considers some of the limitations of possible encoding decoding schemes. Another section that has improved greatly is the explanation of the reasoning for introducing a dynamic scale model. We appreciate the notation changes to denote expected values with the more commonly used notation in statistics. Also, the added magnitude correlation helps explaining the shift to independence. is there any way this can be linked to footnote 3?

*Reviewer #3 (Recommendations for the authors):*

The authors have replied and addressed my (minor) suggestions. The main one was an issue in Figure 5, which has been addressed with a new quantification and plot that I find helpful. The revisions address also the comments from other reviewers, as far as I can tell, and improve the clarity of the paper.

*I have no remaining concerns.*

---

## [Author Response]

Reviewer #1 (Recommendations for the authors):Strengths:A systematic treatment of the properties of natural motion is needed. The point tracking approach used here is a nice approach to resolve many of the challenges facing other approaches to this problem. The evidence that motion statistics are non-Gaussian is clear, and the introduction of the nonlinear scaling parameter to capture that non-Gaussian behavior provides a quite useful characterization of that non-Gaussian behavior.Weaknesses:1. Intuitively, it seems expected that horizontal and vertical velocities should be correlated given that most motion will occur in a direction that is not purely along one of those axes. That intuitive description seems to capture the key features illustrated in Figure 2. Can this physical aspect of the problem be incorporated into the analysis? At a minimum more could be done to develop this intuition, and then use that to intuition to help interpret the more quantitative analyses.

While it is true that most motion samples are not purely along the horizontal or vertical axes in our data, this does not imply an overall correlation between the horizontal and vertical components of the velocity. Correlation between velocity components means that there is some dominant axis of the motion distribution that is not aligned with the horizontal or vertical axes. Some scenes did indeed have small correlations between velocity components, as indicated by a tilt in their raw histograms (not shown). These small correlations were removed as described in the last paragraph of the “Point tracking” section of “Materials and methods.” Thus, horizontal and vertical velocity are linearly uncorrelated in all scenes.

It seems possible the reviewer is referring to the horizontal and vertical speed, that is, the absolute values of the velocity components, and the fact that the mean of the product of the two speeds will be positive. This mean absolute product is not a meaningful measure of correlation, since it will be positive even if the two velocities are independent. After subtracting off the mean speeds to properly calculate the correlation, it will be zero if the velocity components are independent.

This brings up the interesting point that the component speed correlations are, in fact, nonzero for our data, while the component velocity correlations are zero. This is a simple way of demonstrating the nonlinear dependence between velocity components. We have added a simple calculation to the text when first describing heavy-tailed joint distributions (second paragraph of “Heavy-tailed statistics of natural motion”) and describe how this arises in a model with a shared scale parameter (footnote 3 in the same section). We have also added this calculation as a function of time lag in Figure 5C, since it very clearly demonstrates that the nonlinear dependency decays with time lag, which (as other reviewers point out) is not so easily appreciated from the raw data in Figure 5D.

We have also added the following text to avoid confusion about what is meant by velocity and correlation:

“The focus of our analysis is the point velocity, or difference in point positions between subsequent frames: a two-dimensional vector quantity measured in raw units of pixels/frame.” (3rd paragraph of “Results”)

“The lack of tilt in the histogram indicates that the two velocity components are uncorrelated (in this context, correlation between velocity components would indicate a tendency for objects within a scene to move along some diagonal axis relative to the camera).” (2nd paragraph of “Heavy-tailed statistics of natural motion”)

2. Some of the assumptions in the "Coding implications of heavy tails" section were not made sufficiently clear. First, I think that a nonlinearity that mapped the distribution to a Gaussian would introduce signal dependent noise only if the noise occurs prior to the nonlinearity. If that is correct, it should be stated as part of that analysis. Second, doesn't the "demodulation" approach introduced on line 219 also introduce signal dependent noise if the noise is introduced prior to the normalization step? This would seem to violate the conditions under which the simple Gaussian channel argument holds (line 199-201). Because of these concerns it was not clear what to take away from this section.

We have expanded upon the motivations and assumptions for the Gaussian channel argument, which is intended as a minimal, intuitive explanation of the information theoretic implications of our findings. A Gaussian channel introduces additive noise to an encoded input signal, placing the noise after the encoding. The output of the Gaussian channel is then decoded, and the effects of the added noise must be ameliorated by the choice of the input encoding function. In the “Gaussianization” example, the encoding and decoding are done by the nonlinear function f and its inverse, respectively. We have clarified that it is the noise associated with the decoded variable that is signal-dependent to avoid confusion with the channel noise, which is, of course, independent. In the “demodulation” example, the decoded variable is also corrupted by signal-dependent noise.

3. Figure 5A and B are described (first paragraph of the dynamics section) as providing evidence for a contribution of a nonlinear scale factor to motion correlations. I am not clear on why that is needed. Both of these would seem to be true for simple linear correlations. I may well be missing something, but if so the argument in the text about the need for the nonlinear scaling in this case should be made more clearly. This concern made it difficult to evaluate the rest of that section, as it was not clear why the scaling was needed. This figure and the associated analysis could be described more clearly, perhaps showing the bow-tie structures from earlier figures.

We have revised the first paragraph of this section to make it clearer, with reference to Figures 2B and 2D, that these are essentially the same observations as before, with the addition of a tilt due to linear correlation, and hence the same conclusions hold. The two forms of dependency coexist. The scale variable is “needed” for the same reason as in the static case, namely to model the heavy tails. The subtle distinction, which we build up to, is that there is not a single scale variable modulating the velocity at different time points, but two correlated scale variables. These empirical observations all serve to motivate the full ARGSM model.

Line 206: I believe the reference here should be to Figure 4G

Corrected.

Reviewer #2 (Recommendations for the authors):Summary of work:The paper introduces a model of motion in video sequences from natural scenes. The main contribution of the paper is to show that the statistics of the velocity of the motion of key points in natural scenes are well captured by Gaussian scale mixtures (GSM) models. Two hypotheses for this are that: (1) the distance of the moving objects to the camera control the range of angular motion; and (2) the scale of the driving forces that create motion may fluctuate over time. The proposed model introduces a GSM process which is a combination of an underlying Gaussian process and a scaling process.Strengths:- The proposed model elegantly captures the statistics of the velocity distribution for vertical and horizontal direction in the video sequence.- The authors provide an intuitive information theoretic justification for the normalization process that involves reversing the scaling operation to recover the underlying Gaussian latent factor of their model.- The justification of the model's complexity and the gains in predictive accuracy using the Akaike Information Criterion (AIC) add strength to the argument.- The choice of the distribution of the scaling variable as a $\log$ normal random variable provides an easy way to track model in terms of relevant statistics like variance and kurtosis that apply to testing fits and justifying the coding efficiency arguments.- The introduction of a dynamic GSM using Gaussian processes for both the underlying Gaussian variable and the $\log$ of the scaling factors is also well justified with comparison to other alternative models with static scaling or time independence.Weaknesses:- While the model using $\log$ normal scaling variables provides tractable statistics it may not be the most accurate and also not fully justified from the underlying phenomenon (at least this point is not made clear in the paper)

This is a valid point. Different authors in the GSM literature use different scale variable distributions, with little consensus on what distribution is best. Vanderstraeten and Beck (2008) (referenced in the main text) argue that a log-normal distribution arises from a maximum entropy argument in some systems in statistical physics (along with γ and inverse γ, depending on the problem). However, we are operating quite far from such a first principles argument in our own work. The log-normal GSM empirically fits the marginal distributions quite well. To capture the time-dependency of the scale, it is very helpful to model temporal correlations with an underlying Gaussian random variable, then pass this through a rectifying nonlinearity to get a positive scale variable. Other nonlinearities could be used, but the exponential function is a simple and natural choice. Thus, the log-normal distribution both fits the data well and is simple to work with. Other distributions have the advantage of giving closed-form expressions for the resulting GSM distribution, but this turns out not to be useful for estimating the multivariate and time-varying models.

- The proposed model discusses a way to represent velocities as a multiplicative mix. It would benefit from making more explicit connection between the GSM model's representation and areas in the brain that may represent velocities. If the work in this area is lacking, does the proposed model hint at areas that could be encoding velocities?

Many parts of the brain encode velocity, from simple direction-selective cells in the retinas of many mammals to highly refined representation in area MT of the primate cortex. We do not believe that our GSM model of motion points to particular brain regions, but rather highlights the problem of an underlying fluctuating scale that should apply to all velocity-encoding cells. Mechanisms for dealing with this problem range from single-cell adaptation mechanisms to circuit-level normalization, as mentioned in the Discussion.

Reviewer #3 (Recommendations for the authors):This is one of the most enjoyable manuscripts I have reviewed in a while! The topic of object motion estimation and tracking is central to visual and visuomotor processing. While this is a very broad topic, the manuscript focuses on a small, well-defined part of it and reports a study with exemplary execution. The main finding is that the statistics of motion of points (a more tractable proxy for "objects") in natural videos are heavy-tailed, highly predictable, and well modeled with Gaussian Scale Mixtures and their new dynamic (auto-regressive) extensions. The main implications are that these regularities should be exploited by the visual system to predict object motion and by neurons to adapt to object motion statistics across different environments, leading to several experimental predictions.The authors propose to characterize the statistics of local motion over time in natural videos, and to develop generative models that capture those statistics. To do so, they choose a database of natural videos with moderate complexity, including no camera motion and closeup scenes of uninterrupted dense motion (e.g. water flowing or swarms of insects). This allows them to use effectively a standard point-tracking algorithm to define the point motion trajectories for analysis, and to restrict their analysis to 1-second-long clips of uninterrupted but substantially variable motion. They explain well the importance of studying trajectories (which are dynamic motion features) rather than simply motion energy or optic flow, which are in a sense instantaneous motion features.Their analysis provides compelling evidence that both the instantaneous and dynamic statistic of point velocities deviate from Gaussian, have long tails, and may result from a Gaussian-distributed velocities multiplied by a global scaling factor, leading them to formulate Gaussian Scale Mixture models (GSM). GSMs have been used to model statistics of images (some references cited) and instantaneous statistics of movies (some references missing), but not in the context of object/point motion and not considering entire trajectories, as they do here. The analysis carefully quantifies the improvement in capturing motion statistics with their models over simpler alternatives.

We thank this reviewer for their enthusiasm and for making a number of helpful recommendations.

[Editors’ note: what follows is the authors’ response to the second round of review.]

Reviewer #1 (Recommendations for the authors):This is a revision of a paper describing the statistics of natural motion. This is an issue that, to my knowledge, has not been described in detail and the work in the paper fills this gap nicely. The revisions have improved the paper substantially. I have a few remaining suggestions.

We appreciate your attention to detail! Here are our changes:

Line 117: define "good trajectories"

We have changed this to “the ensemble of trajectories” to avoid subjective language.

Line 130-132: the heavy tails are clear in Figure 2 but not in Figure 1. Can you refer ahead here to help a reader out?

We have added the parenthetical “(see \FIG{Figure 2}A for a log scale plot that emphasizes the tail behavior).”

Line 207 (and below): doesn't this strategy often also increase noise since it must be based on an estimate of X (with noise) and the estimate of S could be small in some instances?

In the limit that no channel noise is added to the estimate of *S*, the scaling can be inverted perfectly regardless of its value, so errors in estimating *S* do not increase noise. They do, however, make the distribution of the estimate of *Y* less Gaussian and therefore decrease channel efficiency.

Line 286: define "innovation noise" – particularly why it is given that name

This is jargon from the signal processing literature. We have replaced all instances with “driving noise,” which should be more intuitive for a broad audience.

Line 287: why is the innovation noise small?

This is an empirical observation. We have replaced the relevant sentence with, “Since $\σ_\zeta^2$ is small for the estimated models, indicating that the scale fluctuations are highly predictable at the level of single time steps, we expect this term to have little effect.”

Lines 291-295: What is going on with the insect movies?

These do show a faster timescale and motivate recording with a higher framerate camera (outside the scope of this paper, though). We have added the parenthetical “(a consequence of their very short velocity correlation times).”

Lines 291-295: How can the variance explained by the naive model R^2_X be higher than that for the more complete model?

We have double-checked the numerics and this appears to be just some quirk of the model fitting. It is unexpected but not mathematically impossible, since the ARGSM model has a more complicated likelihood function and fitting procedure. The AIC demonstrates that ARGSM is a better fit than AR in terms of likelihood, and it is likelihood that we are optimizing, not R^2. We have added the word “Predictive” to the caption for Figure 7J so that this is less easily confused with a measure of model fit. Since our conclusion is that AR and ARGSM are quite similar in terms of R^2, we don’t feel it is necessary to highlight this beyond pointing it out parenthetically as we do now.

Lines 302-304: the prediction section ends on a technical note. It would be helpful to have a summary of the main take home point from that section.

We have made this technical sentence a footnote and added the concluding sentence, “The results of this section indicate that this is feasible with or without taking the fluctuating scale into account.”

Reviewer #2 (Recommendations for the authors):After reading the revised version of the manuscript, I can see that the concerns raised by all reviewers where either incorporated into the manuscript or clarified in the author responses. The section on the coding implications of heavy tails presents the argument better as it also considers some of the limitations of possible encoding decoding schemes. Another section that has improved greatly is the explanation of the reasoning for introducing a dynamic scale model. We appreciate the notation changes to denote expected values with the more commonly used notation in statistics. Also, the added magnitude correlation helps explaining the shift to independence. is there any way this can be linked to footnote 3?

Thank you for your response and helpful suggestion. We have added the phrase “discussed above (see footnote 3)” to the text introducing Figure 5C.